# The KASH5 protein involved in meiotic chromosomal movements is a novel dynein activating adaptor

Ritvija Agrawal[1], John P Gillies[1], Juliana L Zang[1], Jingjing Zhang[2], Sharon R Garrott[1,3], Hiroki Shibuya[2]*, Jayakrishnan Nandakumar[1]*, Morgan E DeSantis[1,3]*

[1]Department of Molecular, Cellular and Developmental Biology, University of Michigan-Ann Arbor, Ann Arbor, United States; [2]Department of Chemistry and Molecular Biology, University of Gothenburg, Gothenburg, Sweden; [3]Biological Chemistry, University of Michigan, Ann Arbor, United States

**Abstract** Dynein harnesses ATP hydrolysis to move cargo on microtubules in multiple biological contexts. Dynein meets a unique challenge in meiosis by moving chromosomes tethered to the nuclear envelope to facilitate homolog pairing essential for gametogenesis. Though processive dynein motility requires binding to an activating adaptor, the identity of the activating adaptor required for dynein to move meiotic chromosomes is unknown. We show that the meiosis-specific nuclear-envelope protein KASH5 is a dynein activating adaptor: KASH5 directly binds dynein using a mechanism conserved among activating adaptors and converts dynein into a processive motor. We map the dynein-binding surface of KASH5, identifying mutations that abrogate dynein binding in vitro and disrupt recruitment of the dynein machinery to the nuclear envelope in cultured cells and mouse spermatocytes in vivo. Our study identifies KASH5 as the first transmembrane dynein activating adaptor and provides molecular insights into how it activates dynein during meiosis.

*For correspondence:
hiroki.shibuya@gu.se (HS);
jknanda@umich.edu (JN);
mdesant@umich.edu (MEDeS)

Competing interest: The authors declare that no competing interests exist.

## Editor's evaluation

This manuscript identifies a meiosis-specific protein that recruits and activates the motility of the dynein-1 transport machinery at the nuclear envelope. In prophase I of meiosis, dynein moves chromosomes tethered to the nuclear envelope to expedite the search and pairing between homologous chromosomes. Previous studies have shown that dynein tethers to chromosomes via the LINC complex, which consists of a SUN protein and transmembrane KASH protein. KASH5 comprises all the known features of bona fide cargo adaptors of dynein, and this manuscript demonstrated that KASH5 directly binds dynein and activates its processive motility.

## Introduction

ytoplasmic dynein-1 (dynein) is the primary retrograde, microtubule-associated molecular motor in most eukaryotes (*Cianfrocco et al., 2015*; *Schroer et al., 1989*). Dynein performs cellular work by coupling the energy derived from ATP hydrolysis to the movement of cellular cargo along the microtubule filament (*Roberts et al., 2013*; *Carter, 2013*; *Carter et al., 2016*). Dynein is a multisubunit protein complex that comprises two copies of six different subunits, including the heavy chain (HC) that contains the ATPase motor, the intermediate chain, the light-intermediate chain (LIC), and three different light chains (*Canty et al., 2021*; *Figure 1—figure supplement 1A*). Dynein is not a processive motor on its own. Dynein achieves processive and directional motility by assembling into

the activated dynein complex that, in addition to dynein, includes the 23-subunit protein complex dynactin and one of a class of proteins called activating adaptors (*McKenney et al., 2014*; *Schlager et al., 2014*; *Urnavicius et al., 2018*; *Grotjahn et al., 2018*; *Urnavicius et al., 2015*; *Figure 1—figure supplement 1A*).

There are ~12 confirmed activating adaptors, defined by their ability to bind dynein and induce fast and processive dynein motility on immobilized microtubules in vitro (*Reck-Peterson et al., 2018*; *Olenick and Holzbaur, 2019*). Each activating adaptor contains an N-terminal domain that binds to the dynein LIC, followed by a coiled-coil domain (CC) that binds to both dynein HC and dynactin (*Reck-Peterson et al., 2018*; *Olenick and Holzbaur, 2019*; *Figure 1A*). In addition to binding dynein-dynactin, each activating adaptor uses its C-terminal domain to bind cellular cargo and link it to the dynein motor (*Reck-Peterson et al., 2018*; *Olenick and Holzbaur, 2019*; *Redwine et al., 2017*). The currently known activating adaptors fall into three families based on the structure of their LIC-binding N-terminal regions (*Lee et al., 2018*; *Lee et al., 2020*; *Schroeder et al., 2014*). LIC-binding domains are either 'CC-boxes' (as in BICDR1 and BicD2) (*McKenney et al., 2014*; *Schlager et al., 2014*; *Urnavicius et al., 2018*; *Elshenawy et al., 2019*), Hook-domains (as in Hook1 and Hook3) (*McKenney et al., 2014*; *Urnavicius et al., 2018*; *Olenick et al., 2016*; *Schroeder and Vale, 2016*), or EF-hand-pairs (as in ninein, ninein-like, and CRACR2a) (*Redwine et al., 2017*; *Lee et al., 2020*; *Wang et al., 2019*). An EF-hand in its canonical form is a $Ca^{2+}$-binding structural motif, although several EF-hand proteins adopt this structure without the aid of $Ca^{2+}$ *Kawasaki et al., 1998*. Consistent with this, some dynein activating adaptors like CRACR2A bind $Ca^{2+}$ and require it for dynein activation, while others like Rab45 and ninein-like do not (*Lee et al., 2020*; *Wang et al., 2019*).

As the primary motor used to traffic all cellular cargo toward the microtubule minus-end (typically clustered near the nucleus), dynein must traffic hundreds of types of cargo, including, but not limited to, membrane-bound vesicles, organelles, RNA-protein complexes, lipid droplets, and viruses (*Reck-Peterson et al., 2018*). How cargo specificity is defined in different biological contexts and for myriad cargo is not well understood, but dynein activating adaptors likely play an important role in this regard. Perhaps some of the most unique cargos trafficked by dynein are chromosomes in prophase I of meiosis. Meiosis involves one round of DNA replication followed by two rounds of cell division to generate haploid gametes (*Zickler and Kleckner, 2015*). Crossover, which is the product of homologous recombination between homologous chromosomes in prophase I, ensures proper segregation of chromosomes and promotes genetic diversity. Dynein drives chromosomal movements to facilitate homolog pairing and meiotic recombination (*Wynne et al., 2012*; *Lee et al., 2015*). These dynein-driven chromosomal movements increase the search space that homologous chromosomes explore in the nucleus to untangle mispaired nonhomologous chromosomes and facilitate proper pairing between homologous chromosomes (*Figure 1B*; *Shibuya and Watanabe, 2014c*).

The nuclear envelope (NE) is intact during prophase I, implying that dynein is separated from its cargo by two lipid bilayers (the inner nuclear membrane [INM] and outer nuclear membrane [ONM]) and the perinuclear space between them (*Figure 1B*). In this context, dynein tethers to chromosomes via the highly conserved linker of nucleoskeleton and cytoskeleton (LINC) complex at the NE (*Meier, 2016*). The LINC complex consists of a SUN protein and a KASH protein that span the INM and ONM, respectively and bind each other in the perinuclear space (*Hieda, 2017*; *Figure 1B*). Mammals encode multiple SUN and KASH proteins that bind to their LINC partner using highly conserved SUN and KASH domains, respectively. Cytosolic dynein immunoprecipitates with KASH5, a KASH protein that is expressed exclusively in prophase I of meiosis and essential for mouse fertility (*Morimoto et al., 2012*; *Horn et al., 2013*). A short ~20 amino acid KASH peptide at the very C-terminus of KASH5 interacts with the SUN protein, SUN1, in the perinuclear space (*Figure 1B* and domain diagram of KASH5-FL in *Figure 1C*). Inside the nucleus, SUN1 binds telomeres at the ends of chromosomes, completing the attachment between dynein and the chromosomal cargo (*Ding et al., 2007*). Despite being a process essential to meiosis progression and fertility, how dynein is activated to drive these movements across the NE remains unknown.

KASH5 protein consists of an N-terminal EF-hand pair followed by a CC domain, reminiscent of the domain architecture of EF-hand pair-containing dynein activating adaptors (*Figure 1C* and *Figure 1—figure supplement 1B*). Here, we tested the hypothesis that KASH5 is a dynein activating adaptor that activates dynein specifically during prophase I of meiosis. We show that KASH5 binds dynein LIC in a manner comparable to other known dynein activating adaptors. Using human cell lysates

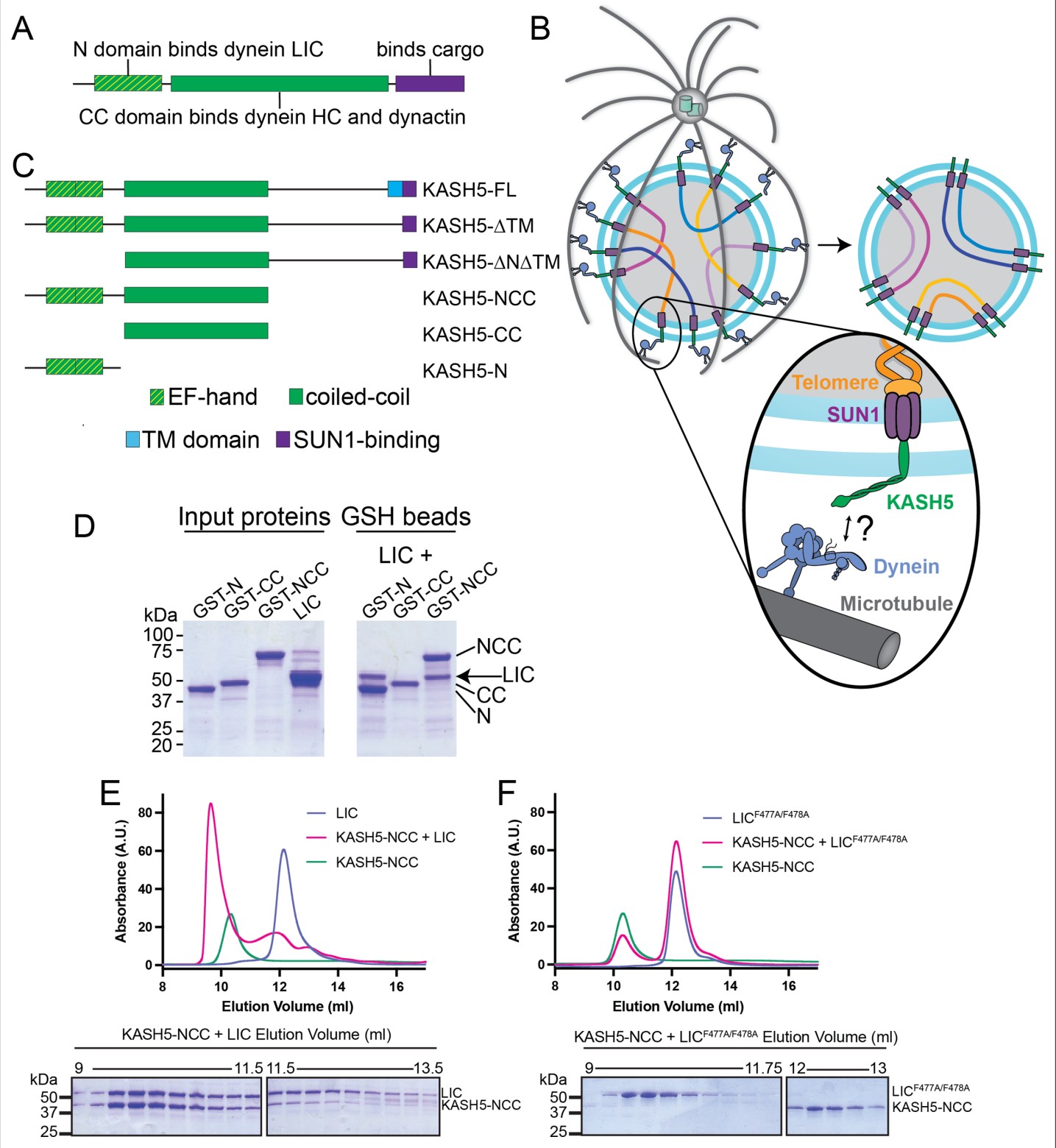

**Figure 1.** KASH5 binds dynein via a direct NCC-light-intermediate chain (LIC) interaction. (**A**) Domain arrangement of a typical dynein activating adaptor showing an N-terminal LIC-interacting domain, a central coiled-coil (CC), and a C-terminal cargo-binding domain. Note that this description is simplified for BicD2 as it is primarily CC and thus uses a CC region toward the N-terminus to bind LIC. (**B**) Schematic for how the telomere and dynein tether to each other at the nuclear envelope with the help of the SUN1-KASH5 complex to move chromosomes and facilitate homolog pairing during meiosis. (**C**) Domain diagram of human KASH5 FL and domain deletion constructs used in this study. (**D**) Pull down of purified proteins on glutathione

*Figure 1 continued on next page*

*Figure 1 continued*

(GSH) beads. Indicated glutathione S-trasferase (GST)-tagged KASH5 constructs were incubated with dynein LIC and pulled down on GSH-beads followed by visualization on an SDS-PAGE using Coomassie-blue staining. Number of replicates, n=2. (**E and F**) $UV_{280}$ absorbance profile (top) and Coomassie-blue staining analysis (bottom) of size-exclusion chromatography (SEC) of KASH5-NCC alone, LIC alone, and a mixture of KASH5-NCC and LIC using either wild type LIC (E) or the LIC F447A/F448A double mutant (F). n represents number of replicates for each SEC run. n=3, 3, 2, 2, and 2 for KASH5-NCC alone, LIC alone, LIC F447A/F448A alone, KASH5-NCC with LIC, and KASH5-NCC with LIC F447A/F448A, respectively.

The online version of this article includes the following source data and figure supplement(s) for figure 1:

**Source data 1.** Unedited SDS-PAGE gels relating to *Figure 1D–F*.

**Figure supplement 1.** KASH5 uses its N-terminal domain and coiled-coil (CC) domain to bind light-intermediate chain (LIC).

**Figure supplement 1—source data 1.** Unedited blots related to *Figure 1—figure supplement 1C*.

**Figure supplement 1—source data 2.** Unedited SDS-PAGE gels related to *Figure 1—figure supplement 1D*.

and purified recombinant protein, we demonstrate that KASH5 is a bona fide activating adaptor that converts dynein and dynactin into a processive motile complex. We identify specific residues in the KASH5 EF-hand pair that mediate the interaction with dynein LIC and demonstrate that disruption of the KASH5-LIC interaction impairs dynein recruitment to the NE of both KASH5-transfected HeLa cells and mouse spermatocytes undergoing meiosis. Our findings establish KASH5 as a novel EF-hand pair-type activating adaptor, the first identified transmembrane (TM) dynein activating adaptor, and the dynein activator responsible for moving chromosomes in meiotic prophase I.

## Results

### KASH5 binds dynein's LIC using a mechanism conserved in other activating adaptors

To determine the regions of KASH5 that promote binding to dynein and dynactin, we generated 3×-FLAG-tagged KASH5-FL (wild type [WT]) and domain deletion constructs lacking one or more of its domains, namely the TM domain, EF-hand pair-containing N-terminal domain (N), CC domain, and the unstructured region between its CC and TM (*Figure 1C* and Methods). Deletion of TM greatly increased the level of soluble KASH5 compared to KASH5 FL (compare lanes five and six in Input, *Figure 1—figure supplement 1C*), allowing us to define a stable, TM-less construct of KASH5 for subsequent biochemical analysis in buffers lacking harsh detergents. Indeed, KASH5-ΔTM co-immunoprecipitates with dynein's HC and LIC and dynactin's p150 subunit (*Figure 1—figure supplement 1C*), consistent with the association of the dynein transport machinery and KASH5 reported previously (*Horn et al., 2013*). KASH5-NCC, and to a lesser extent, KASH5-N, retained association with dynein and dynactin, suggesting that KASH5-NCC recapitulates the entire dynein-interaction interface of KASH5 as suggested previously (*Horn et al., 2013*). Co-immunoprecipitation (Co-IP) data strongly suggest a direct interaction between KASH5 and dynein, but this has not been demonstrated with purified components. We hypothesized that the KASH5 N-terminus, like in other dynein activating adaptors, directly binds dynein LIC (*Figure 1A and C*). We performed glutathione (GSH) bead pulldown of purified GST-fusions of human KASH5 N, CC, and NCC with full-length, untagged human dynein LIC1 (hereafter referred to as LIC). Consistent with co-IP analysis, KASH5 N and NCC, but not CC, bind directly to LIC (*Figure 1D*). Together, our results show that KASH5 directly binds the dynein transport machinery using its EF-hand pair-containing N domain and CC domain.

To determine if KASH5 binds to dynein like other EF-hand pair activating adaptors (*Figure 1—figure supplement 1B*), we incubated purified, untagged KASH5-NCC and LIC (*Figure 1—figure supplement 1D*) and performed size-exclusion chromatography (SEC). Indeed, KASH5-NCC co-eluted with LIC in SEC as a peak distinguishable from KASH5-NCC or LIC alone (*Figure 1E*). Dynein activating adaptors tested to date bind dynein's LIC via a well-conserved pair of consecutive phenylalanine residues in LIC's helix 1 (F447 and F448 in the human LIC1 sequence) (*Figure 1—figure supplement 1E*; *Lee et al., 2018*; *Lee et al., 2020*; *Celestino et al., 2019*). LIC[F447A/F448A], which has both key phenylalanine residues mutated to alanine, disrupts binding to BicD2, Hook3, and CRACR2a (*Lee et al., 2018*; *Lee et al., 2020*). KASH5-NCC and LIC[F447A/F448A] failed to co-elute (*Figure 1F*), consistent with KASH5 using a structural mechanism shared with other dynein activating adaptors to bind dynein LIC.

## KASH5 binds dynein LIC with an affinity comparable to other activating adaptors but with unique stoichiometry

To determine the stoichiometry of the KASH5-LIC interaction, we used SEC coupled to multi-angle light scattering (MALS) to determine the molecular weight of the NCC:LIC complex. NCC:LIC forms a homogenous complex with an experimentally determined molecular weight of 139 kDa, consistent with a 2:1 NCC:LIC stoichiometry (theoretical molecular weight (MW) of 133.5 kDa) (*Figure 2A*). The 2:1 stoichiometry was also observed at threefold higher protein concentrations (~10-fold greater than the $K_d$ determined with isothermal titration calorimetry [ITC]; see below) and despite adding a twofold molar excess of LIC over KASH5-NCC, suggesting that it is not a result of incomplete binding or dissociation of the complex (*Figure 2—figure supplement 1A*). SEC-MALS analysis of NCC (alone) is consistent with a homodimer, defining a quaternary structure for KASH5 that is conserved among other dynein activating adaptors.

The 2:1 stoichiometry suggests that CC dimerization allows two KASH5's EF-hand pairs to bind a single copy of LIC. To further explore the role that KASH5 dimerization plays in LIC binding, we performed SEC-MALS with two different constructs of KASH5-N: GST-KASH5-N and untagged KASH5-N. Both constructs contain KASH5's EF-hand pair, but only GST-KASH5-N would dimerize through its GST tag. SEC-MALS confirmed that GST-KASH5-N is a dimer and untagged KASH5-N is a monomer (*Figure 2B and C*). Next, we incubated each construct with purified LIC. Both GST-KASH5-N and KASH5-N were capable of binding LIC (*Figure 2B and C*). The experimentally determined mass of the complex formed between KASH5-N and LIC was ~10 kDa lower than the theoretical mass for a 1:1 complex (*Figure 2B*). This lower apparent molecular weight could indicate either partial dissociation of the complex during SEC analysis or partial overlap between the peak of the 1:1 complex and excess LIC in the mixture. To ensure that the observed stoichiometry was not limited by the concentration of the proteins, we repeated the SEC-MALS analysis of the LIC-KASH5-N mixture at threefold higher concentration of each protein (*Figure 2—figure supplement 1B*). Increased protein concentration did not alter the ~1:1 stoichiometry. In contrast to KASH5-N and consistent with what we observed with KASH5-NCC, GST-KASH5-N bound with 2:1 stoichiometry, confirming that dimerization of an EF-hand is not accompanied by association of a second LIC polypeptide (*Figure 2B and C*). Together, these data suggest that dimeric KASH5 EF-hands bind a single copy of LIC.

We employed ITC to determine the affinity between KASH5 and LIC. We titrated a LIC peptide that corresponds to the sequence in helix 1 (amino acids (aa) 433–458) into purified KASH5-NCC (*Lee et al., 2018*; *Lee et al., 2020*; *Celestino et al., 2019*). We observed that KASH5-NCC bound the LIC peptide with a dissociation constant $K_d$ = 4.3 µM, which is comparable to the LIC-binding affinities of other activating adaptors (*Lee et al., 2018*; *Lee et al., 2020*; *Figure 2D*). Some EF-hand pair activating adaptors bind dynein in a $Ca^{2+}$-dependent fashion (*Lee et al., 2020*; *Wang et al., 2019*). To test if $Ca^{2+}$ regulates KASH5 binding to dynein's LIC, we performed ITC in the presence of $CaCl_2$ or EGTA to chelate any $Ca^{2+}$ that co-purified with KASH5-NCC. Dissociation constants were very similar for each condition, suggesting that $Ca^{2+}$ does not regulate the NCC-LIC interaction (*Figure 2—figure supplement 1C, D*). Finally, we used ITC to test if the EF-hand pair of KASH5 binds $Ca^{2+}$. Titration of $Ca^{2+}$ into KASH5-NCC did not result in a change of enthalpy, suggesting that the KASH5 EF-hand pair does not bind $Ca^{2+}$ (*Figure 2—figure supplement 1E*).

## KASH5 activates dynein motility

To determine if KASH5 is a dynein activating adaptor, we appended a C-terminal FLAG and an N-terminal green fluorescent protein (GFP) tag to KASH5-ΔTM in a pcDNA3 backbone for mammalian cell expression (*Redwine et al., 2017*). We expressed these constructs in HEK 293T cells, immunoprecipitated the cellular lysates on anti-FLAG resin, and determined if the GFP-tagged KASH5 constructs displayed processive motility on microtubules using total internal reflection fluorescence microscopy (TIRF) (*Figure 3A*). Observed processive motility would indicate that KASH5 co-precipitated the activated dynein complex. Robust motility was observed with the positive control (GFP-BicD2-FLAG), while very little motility was observed with the negative control (GFP-FLAG) (*Figure 3B and C* and *Figure 3—figure supplement 1A-C*). GFP-KASH5-ΔTM-FLAG and GFP-KASH5-NCC-FLAG displayed processive movements on microtubules (*Figure 3B and C*). We quantified the velocity, run length, percentage of processive events, and the landing rate of dynein on microtubules in the presence of BicD2 and the KASH5 constructs. Dynein associated with GFP-KASH5-ΔTM-FLAG, GFP-KASH5-NCC-FLAG,

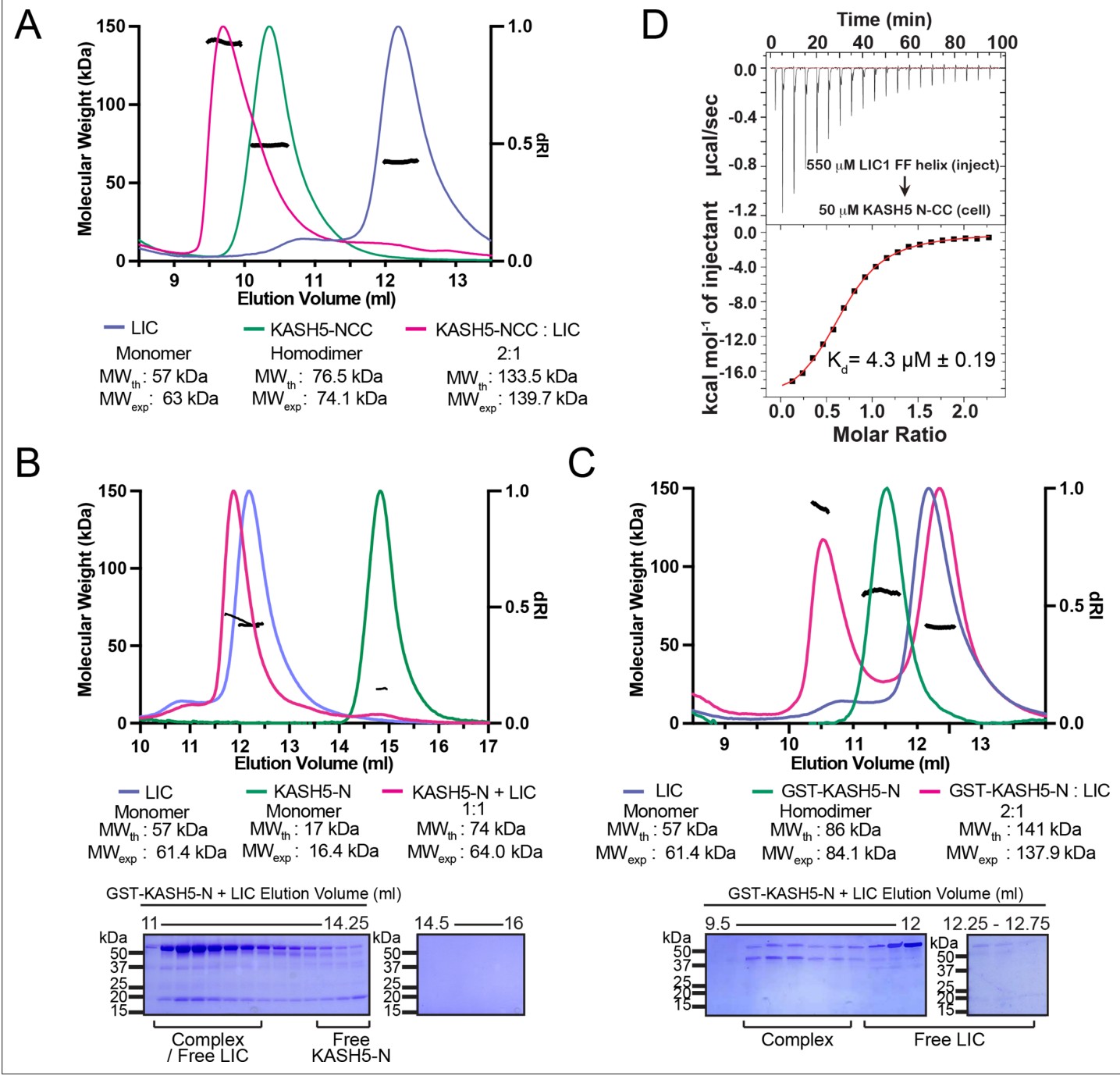

**Figure 2.** KASH5 directly binds dynein light-intermediate chain (LIC) with a 2:1 stoichiometry. (**A**) Size-exclusion chromatography (SEC)-multi-angle light scattering (MALS) analysis of KASH5-NCC alone, LIC alone, and the KASH5-NCC-dynein-LIC complex (SEC profile data same as in *Figure 1E*) showing that KASH5-NCC is homodimeric while the NCC-LIC complex adopts a 2:1 stoichiometry. n represents number of replicates for each SEC-MALS run. n=2 each for KASH5-NCC alone, LIC alone, and KASH5-NCC with LIC. (**B and C**) SEC-MALS analysis of KASH5-N alone, LIC alone, and a mixture of KASH5-N and LIC using either untagged KASH5-N (B) or GST-tagged KASH5-N (C). Coomassie-blue stained SDS-PAGE analysis for the indicated KASH5-LIC mixtures is shown below the SEC-MALS profile. n represents number of replicates for each SEC-MALS run. n=1, 1, 3, and 1 for KASH5-N, GST-KASH5-N, KASH5-N with LIC, and GST-KASH5 with LIC, respectively. (**D**) Isothermal titration calorimetry (ITC) analysis of KASH5-NCC with the dynein LIC$^{433-458}$ peptide containing F447 and F448. Mean and SE of the mean of the dissociation constant ($K_d$) are indicated for a biological duplicate.

The online version of this article includes the following source data and figure supplement(s) for figure 2:

**Source data 1.** Unedited SDS-PAGE gels relating to *Figure 2B and C*.

**Figure supplement 1.** KASH5-NCC binds dynein in a 2:1 stoichiometry and in a Ca$^{2+}$-independent manner.

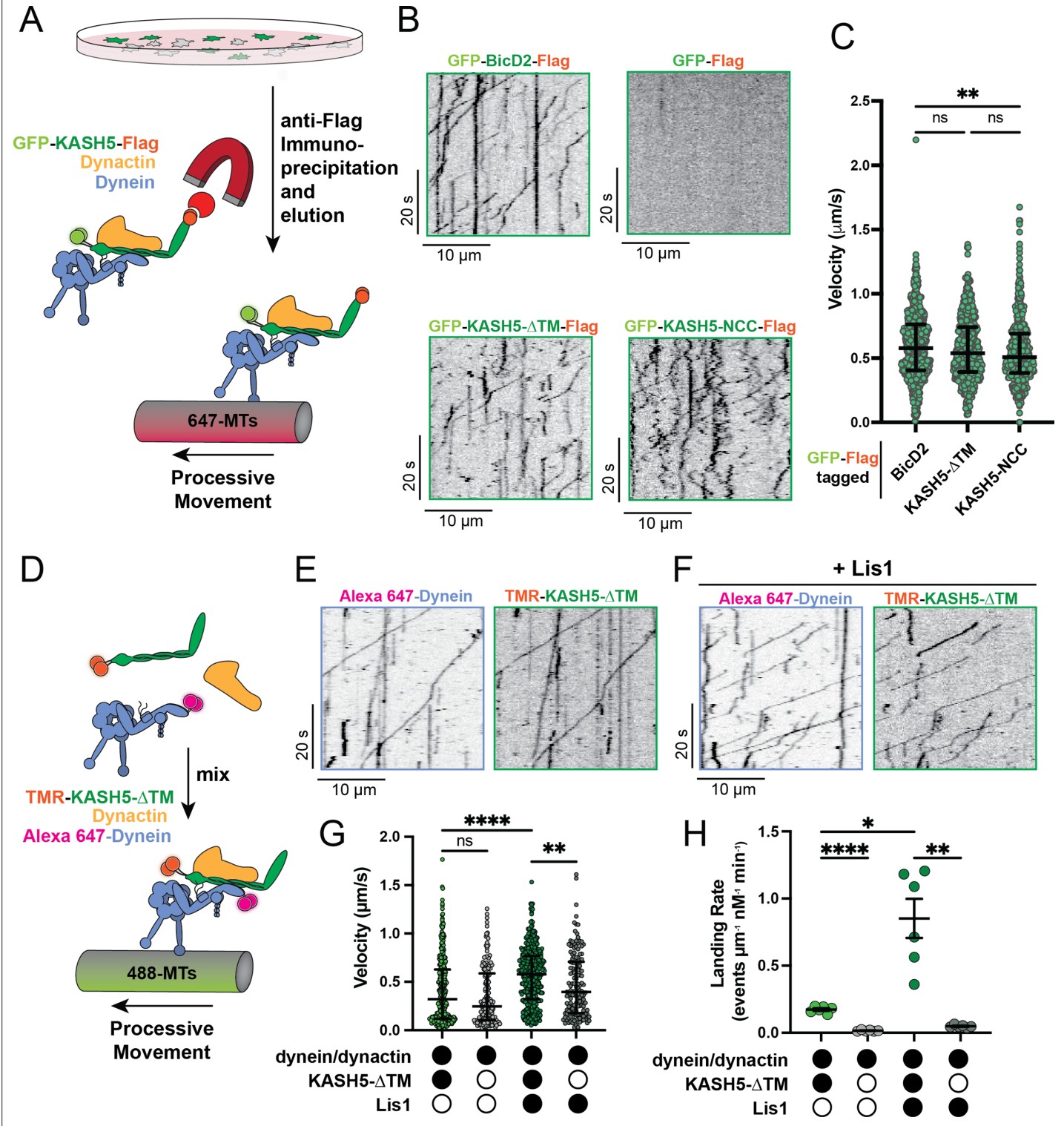

**Figure 3.** KASH5 activates dynein. (**A**) Schematic for the IP-total internal reflection fluorescence (TIRF) assay to measure dynein motility of anti-FLAG immunoprecipitates of GFP-FLAG-tagged KASH5-ΔTM and BicD2. (**B**) Kymographs showing motility of immunoprecipitated complexes containing indicated GFP-FLAG-tagged constructs monitored by GFP fluorescence using TIRF microscopy. (**C**) Velocity of processive events from a total of two movies from two biological replicates (four movies analyzed in total). Each data point represents an individual processive event; n=694, 615, and 484 for BicD2, KASH5-ΔTM, and KASH5-NCC, respectively. Median and interquartile range shown. Significance determined from a Kruskal-Wallis test with Dunn's multiple comparisons test. ns, not significant; **p≤0.01. (**D**) Schematic for the TIRF assay with purified proteins. (**E and F**) Kymographs of KASH5-

*Figure 3 continued on next page*

*Figure 3 continued*

dynein-dynactin complexes monitored by differentially fluorophore-labeled dynein and KASH5 in the absence (E) or presence (F) of Lis1. (**G**) Velocity of processive events from a total of two biological replicates (six movies analyzed in total). Each data point represents an individual processive event; n=398, 414, 174, and 146 for dynein and dynactin with KASH5, KASH5 +Lis1, Buffer, and Lis1, respectively. Median and interquartile range shown. Significance determined from a Kruskal-Wallis test with Dunn's multiple comparisons test. ns, not significant; **p≤0.01; ****p≤0.0001. (**H**) Landing rate for the observed motile events with purified proteins and complexes from a total of two biological replicates (six movies analyzed in total). n values are derived from the average processive events/micron from all microtubules analyzed in a movie; n=6. Mean and SE of the mean shown. Significance determined from a Brown-Forsythe and Welch ANOVA test with Dunnett's T3 multiple comparison test. *p≤0.05; **p≤0.01; ***p≤0.001.

The online version of this article includes the following source data and figure supplement(s) for figure 3:

**Source data 1.** Numerical source data relating to *Figure 3C, G and H*.

**Figure supplement 1.** Extended analysis for total internal reflection fluorescence (TIRF) experiments.

**Figure supplement 1—source data 1.** Numerical source data relating to *Figure 3—figure supplement 1*.

**Figure supplement 1—source data 2.** Unedited SDS-PAGE gels relating *Figure 3—figure supplement 1D*.

**Figure supplement 2.** KASH5 contains a Spindly motif.

and GFP-BicD2-FLAG all moved at velocities ~0.5 μm/s (*Figure 3C*). Interestingly, BicD2 immunoprecipitates displayed slightly enhanced run lengths, processivity, and landing rate compared to either KASH5 construct (*Figure 3—figure supplement 1A-C*). Although the significance of the differences between BicD2- and KASH5-dependent dynein motility is unclear, it may reflect the different biological contexts the two proteins function in. Additionally, KASH5-ΔTM performed slightly better than KASH5-NCC in the immunoprecipitation (IP)-TIRF experiments, suggesting that regions downstream of the KASH5 CC may contribute to dynein motility (*Figure 3C* and *Figure 3—figure supplement 1A-C*). Indeed, C-terminal to the CC in KASH5 is a putative Spindly motif, which in other activating adaptors binds to the pointed end of dynactin and facilitates dynein motility (*Gama et al., 2017*; *Figure 3—figure supplement 2A-C*). Altogether, the IP-TIRF data support the hypothesis that KASH5 is a newly identified dynein activating adaptor.

Activating adaptors are also able to promote dynein motility in a fully reconstituted system (*McKenney et al., 2014*; *Schlager et al., 2014*). To test if KASH5 can activate dynein with only purified components, we purified SNAP-tagged recombinant human dynein from an insect cell expression system and human dynactin from HEK 293T cells stably expressing FLAG-p62, as we have described previously (*Htet et al., 2020*), and recombinant Halo-KASH5-ΔTM from *Escherichia coli* (*Figure 3—figure supplement 1D* and see Methods). We selected KASH5-ΔTM over Halo-tagged KASH5-NCC in this analysis because the former encompasses all soluble domains of KASH5, including the putative Spindly motif (*Figure 1C* and *Figure 3—figure supplement 2A-C*) and showed improved dynein motility in the IP-TIRF analysis (*Figure 3C* and *Figure 3—figure supplement 1A-C*). The purified proteins were assembled in the presence of ATP and an oxygen scavenger system and imaged as they associated with microtubules using TIRF (*Figure 3D*). Dynein was labeled with either tetramethylrhodamine (TMR) or Alexa-647 via the SNAP tag in all experiments. In experiments where dynein was labeled with Alexa-647, we labeled KASH5 with TMR via the Halo tag. Processive dynein motility was observed in the presence of KASH5-ΔTM (*Figure 3E, G and H* and *Figure 3—figure supplement 1E, F*) but not in its absence (*Figure 3G and H*, and *Figure 3—figure supplement 1E-G*). Moving Alexa-647 dynein was almost always colocalized with TMR-labeled KASH5-ΔTM (*Figure 3E*). Together, the TIRF motility data qualify KASH5 as a newly identified dynein activating adaptor, making it one of about a dozen characterized members of this family of dynein regulators.

The velocity of the processive events recorded with purified dynein, dynactin, and KASH5 was significantly slower than in the IP-TIRF motility experiments (median velocity of 0.320 μm/s with purified KASH5-ΔTM versus 0.538 μm/s with dynein immunoprecipitated by KASH5-ΔTM) (*Figure 3C and G*). We reasoned that the reconstitutions with purified proteins were missing a dynein regulatory factor that promotes activity. Lis1 increases the velocity of dynein-dynactin-activating adaptor motility by promoting dynein conformations that drive association with dynactin and an activating adaptor (*Htet et al., 2020*; *Elshenawy et al., 2020*; *Gutierrez et al., 2017*; *Markus et al., 2020*; *Baumbach et al., 2017*; *Qiu et al., 2019*). Inclusion of purified Lis1 in the in vitro reconstituted TIRF experiments resulted in a significant increase in the velocity of motile events (*Figure 3F*). In fact, the velocity of dynein-dynactin-KASH5 processive events in the presence of Lis1 is comparable to that observed in

the IP-TIRF motility experiments (median velocity of 0.577 µm/s). Consistent with observations made with other activating adaptors, Lis1 also increased the landing rate, run length, and percent processivity of dynein-dynactin-KASH5 complexes (*Htet et al., 2020*; *Elshenawy et al., 2020*; *Gutierrez et al., 2017*; *Baumbach et al., 2017*; *Figure 3H* and *Figure 3—figure supplement 1E, F*). These data suggest that Lis1 likely drives the association of dynein-dynactin-KASH5 complexes during meiosis, as has been demonstrated for other activators in the cell (*Htet et al., 2020*; *Qiu et al., 2019*; *Splinter et al., 2012*).

## Specific residues of the KASH5 EF-hand that mediate the interaction with LIC and dynein activation

To identify residues in KASH5 that mediate the interaction with dynein, we generated a homology model of KASH5's EF-hand pair bound to LIC's helix 1 peptide harboring the FF motif based on the structure of CRACR2a EF-hand pair bound to this peptide (*Lee et al., 2020*) (PDB: 6PSD) (*Figure 4A–C*; see Methods). Based on this model, we identified nine conserved amino acids in the KASH5 EF-hand pair that surround the hydrophobic surface of LIC harboring F447 and F448 (*Figure 4C*): I36, T40, Y60, V64, R73, L77, F97, L98, and M101 (*Figure 4A and C*). We substituted each of these residues with aspartate residues (D) in the GFP-KASH5-ΔTM-FLAG background to disrupt the putative hydrophobic interface, expressed each construct in HEK 293T cells, and performed both co-IP and IP-TIRF motility experiments as described in *Figure 1—figure supplement 1C* and *Figure 3A–C*, respectively. All the KASH5 EF-hand mutants tested showed impaired binding to dynein, albeit to varying extents (*Figure 4D and E*). All the mutants co-precipitated at least some motile dynein, as visualized by GFP-KASH5-ΔTM-FLAG in TIRF (*Figure 4F*). Most of the mutants displayed motile events with comparable velocity, processivity, landing rate, and run length as WT KASH5-ΔTM (*Figure 4F–H* and *Figure 4—figure supplement 1A, B*). However, three of the mutants, KASH5-ΔTM$^{L77D}$, KASH5-ΔTM$^{F97D}$, and KASH5-ΔTM$^{M101D}$ displayed severely impaired dynein association and very little processive motility as visualized via TIRF (*Figure 4D–G* and *Figure 4—figure supplement 1A*).

Though all KASH5 variants negatively affected binding to dynein, the velocity of the processive events that were observed for each mutant was not significantly different than KASH5-ΔTM (*Figure 4F and H*). Even for KASH5-ΔTM$^{L77D}$, KASH5-ΔTM$^{F97D}$, and KASH5-ΔTM$^{M101D}$, where we observed very few processive events, the velocity was nearly identical to that induced by KASH5-ΔTM (*Figure 4G and H* and *Figure 4—figure supplement 1A*). This result suggests that the binding between the EF-hand pair in dynein activating adaptors and the LIC promotes the formation of the activated dynein complex but that the activating adaptor-LIC interaction is not required for fast, processive motility on microtubules.

## Disrupting the binding interface between KASH5 and LIC impairs dynein relocalization to the NE in HeLa cells

To assess if the interaction between KASH5 and dynein can recruit dynein to the NE, we first established an immunofluorescence (IF)-based dynein relocalization assay in HeLa cells that stably express dynein HC as a GFP fusion (*Poser et al., 2008*). We transfected these cells with vehicle, SUN1-myc only, or SUN1-myc plus various FLAG-KASH5 constructs to assess if KASH5 can relocalize dynein to the NE. Importantly, there is no endogenous KASH5 in HeLa cells because its expression is limited to prophase I of meiosis. Though there is endogenous SUN1 in HeLa cells, we observed more robust localization of transfected FLAG-KASH5-FL to the nuclear periphery when we co-transfected SUN1-myc (data not shown). We reason that the extent of KASH5 localization at the NE is limited by the SUN1 level, which is sub-stoichiometric to FLAG-KASH5-FL unless co-transfected with it.

As expected, GFP-dynein was relatively diffuse throughout the cytoplasm with no distinct nuclear signal in cells when neither SUN1 nor KASH5 was transiently transfected (*Figure 5A*). In contrast, a discrete ring of GFP-dynein was visually discernible in essentially all cells expressing moderate levels of both SUN1-myc and FLAG-KASH5-FL IF signal, but not in cells expressing SUN1-myc alone or SUN1-myc plus FLAG-KASH5-ΔTM (which itself does not localize to the NE; *Figure 5A*). Dynein enrichment at the NE was quantified by determining the ratio of the mean gray value of GFP-dynein in a 1 µm-thickness ring around the nucleus to the mean gray value of GFP-dynein in the cytoplasm (*Figure 5B*). Indeed, SUN1-myc plus FLAG-KASH5-FL expression caused a significant increase in dynein enrichment at the NE relative to vehicle, SUN1-myc alone, or SUN1-myc plus Flag-KASH5-ΔTM controls (*Figure 5B*). To determine the importance of the KASH5 EF-hand pair-dynein

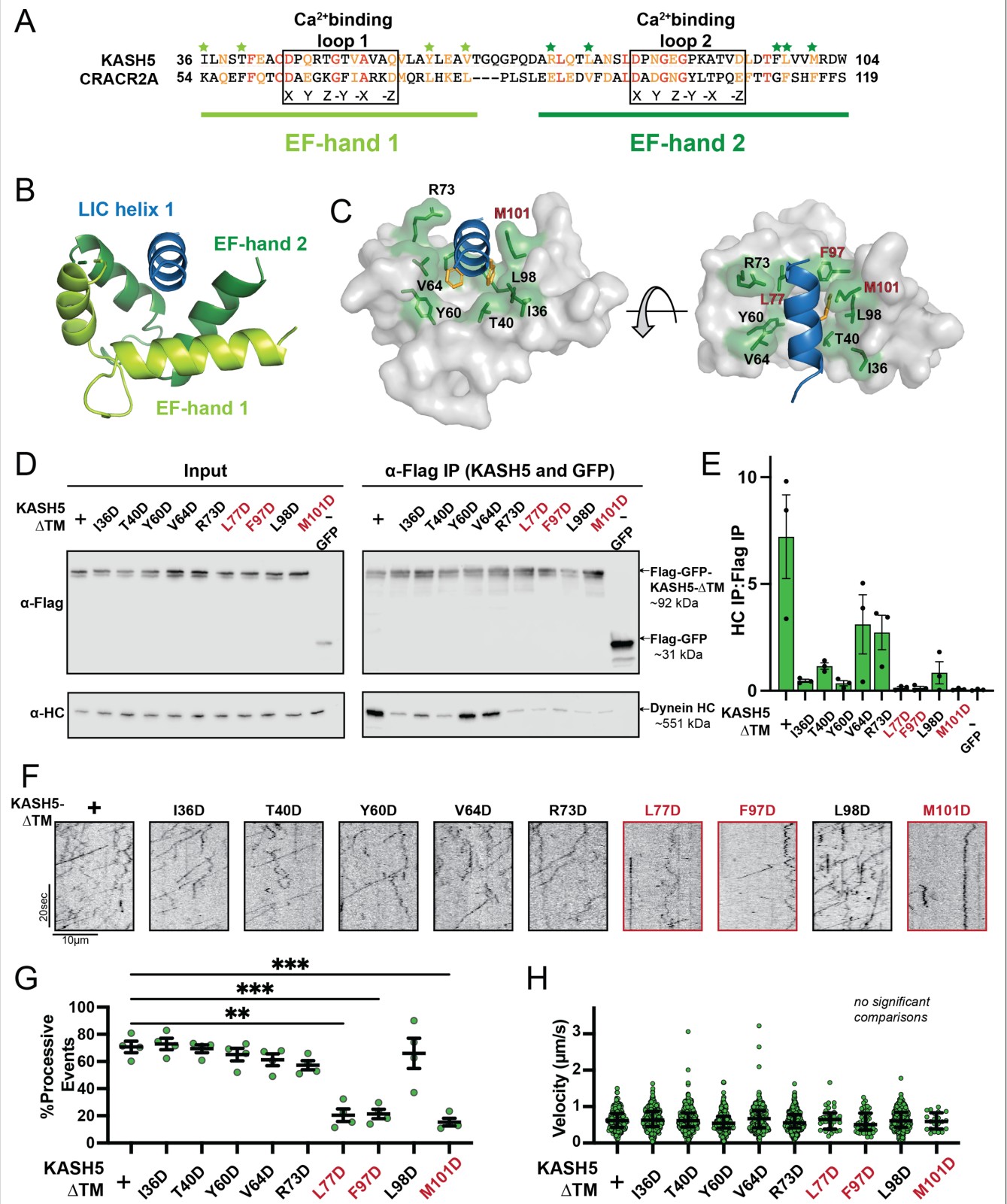

**Figure 4.** Mutations in the EF-hand of KASH5 abrogate association with active dynein complexes in the cell. (**A**) Alignment of the EF-hand pairs of KASH5 and previously characterized dynein activator CRACR2A showing the putative Ca²⁺-binding loop. Conserved and similar residues are shown in red and orange, respectively. Residues that were mutated are indicated with green asterisks. (**B and C**) Homology model of the KASH5 EF-hand pair bound to the dynein light-intermediate chain (LIC) helix 1 with LIC F447/448 shown in orange and the putative KASH5 EF-hand residues mutated in

*Figure 4 continued on next page*

*Figure 4 continued*

this study that form a binding pocket around the helix shown in green. (**D**) Anti-FLAG co-immunoprecipitation (co-IP) analysis of HEK 293T cell lysates containing transiently transfected FLAG-tagged KASH5-ΔTM and indicated mutants; the three mutants with the most drastic binding defect highlighted in red. (**E**) The immunoprecipitation signal for dynein heavy chain (HC) was quantified for each of the samples represented in panel D by dividing the western blot band intensity of HC by that of the FLAG-KASH5-ΔTM band in that lane of the IP fraction. Mean and SE of the mean from a triplicate set of experiments are shown. (**F**) Representative kymographs from motility experiment for each mutant. (**G**) Percent processive events for each mutant. Mean and SE of the mean shown. Percent of processive events from a total of two movies from two biological replicates. n values are derived from the average percent processive events from all microtubules analyzed in a movie; n=4. Significance determined from a Brown-Forsythe and Welch ANOVA test with Dunnett's T3 multiple comparison test. **p≤0.01; ***p≤0.001. Only pairwise comparisons with a p-value p≤0.05 are shown. (**H**) Velocities of processive events for each mutant. Median and interquartile shown. The distribution of velocities for each mutant was compared to KASH5-ΔTM with a Kruskal-Wallis test with Dunn's multiple comparisons test. No pairwise comparison was significantly different. For most mutants, data was obtained from two movies from two biological replicates (four movies analyzed each). For L77D, F97D, and M101D, an additional biological replicate (i.e. an additional two movies) was collected to capture more events. Each data point represents an individual processive event; n=288, 488, 453, 284, 363, 417, 34, 43, 410, and 21 for wild type (WT), I36D, T40D, Y60D, V64D, R73D, L77D, F97D, L98D, and M101D, respectively.

The online version of this article includes the following source data and figure supplement(s) for figure 4:

**Source data 1.** Numerical source data relating to *Figure 4E, G and H*.

**Source data 2.** Unedited blots relating to Figure D.

**Figure supplement 1.** Extended motility analysis of KASH5 mutants.

**Figure supplement 1—source data 1.** Numerical source data relating to (*Figure 4—figure supplement 1A, B*).

LIC interaction for recruiting dynein to the NE, we transfected SUN1-myc plus either FLAG-tagged KASH5-FL$^{L77D}$, KASH5-FL$^{F97D}$, or KASH5-FL$^{M101D}$ and assessed dynein localization to the NE. Very few, if any, GFP-dynein ring structures were visually discernible in cells expressing the three KASH5 mutants (*Figure 5A*). Quantitation of the data showed that expression of both FLAG-tagged KASH5-FL$^{L77D}$ and KASH5-FL$^{F97D}$ significantly reduced the enrichment of GFP-dynein around the nuclear periphery (*Figure 5B*). FLAG-KASH5-FL$^{M101D}$ also seemed to reduce the ratio of GFP-dynein intensity at the NE versus cytoplasm compared to Flag-KASH5-FL, but the magnitude of the phenotype was not statistically significant.

The KASH5-dynein interaction can, in principle, not only cause cytosolic dynein relocalization to the NE but also leach KASH5 away from the NE to dynein in the cytoplasm. As such, mutations that disrupt KASH5's binding with dynein could increase the enrichment of KASH5 at the NE versus the cytoplasm. We observed that all three KASH5 mutant proteins were more enriched at the NE than FLAG-KASH5-FL although the phenotype was not statistically significant for KASH5-FL$^{F97D}$ (*Figure 5C*). Together, our intracellular localization experiments in HeLa cells show that single mutations disrupting binding between KASH5's EF-hand pair and dynein's LIC are sufficient to disrupt dynein recruitment to the NE.

## Disrupting the interaction between KASH5 and LIC inhibits the localization of dynactin at NE-tethered telomeres in mouse spermatocytes

KASH5 acts specifically in meiotic prophase I to facilitate telomere motility along the NE (*Morimoto et al., 2012*; *Horn et al., 2013*). We hypothesized that exogenous expression of dynein-binding-deficient mutants of KASH5 would outcompete endogenous KASH5 at the NE to reduce dynein recruitment to telomeres in mouse spermatocytes. To test this, we electroporated mouse KASH5 mutant proteins, L147D, F167D, and M171D (corresponding to L77D, F97D, and M101D in human KASH5, respectively) as GFP fusions into WT murine spermatocytes. Full-length KASH5 and all three mutants localize to meiotic telomeres, confirming that the mutations do not perturb binding of KASH5 to SUN1 or SUN1 to the telomeres (*Morimoto et al., 2012*; *Figure 6A*). However, the staining of endogenous dynactin subunit p150 at telomeres was significantly decreased in spermatocytes expressing GFP-KASH5$^{L147D}$ (the telomere-localized p150 signal intensity in the mutant-expressing cells was 73% that of cells not expressing any GFP-tagged protein), suggesting that a single mutation in the KASH5-dynein interface is sufficient to disrupt the localization of the dynein transport machinery in a dominant fashion in WT spermatocytes (*Figure 6A and B*). The GFP-KASH5$^{F167D}$ and GFP-KASH5$^{M171D}$ mutants did not cause dominant phenotypes, although they are expressed (as assessed by western blot; *Figure 6C*) and enriched at telomeres (as assessed by GFP foci intensity at

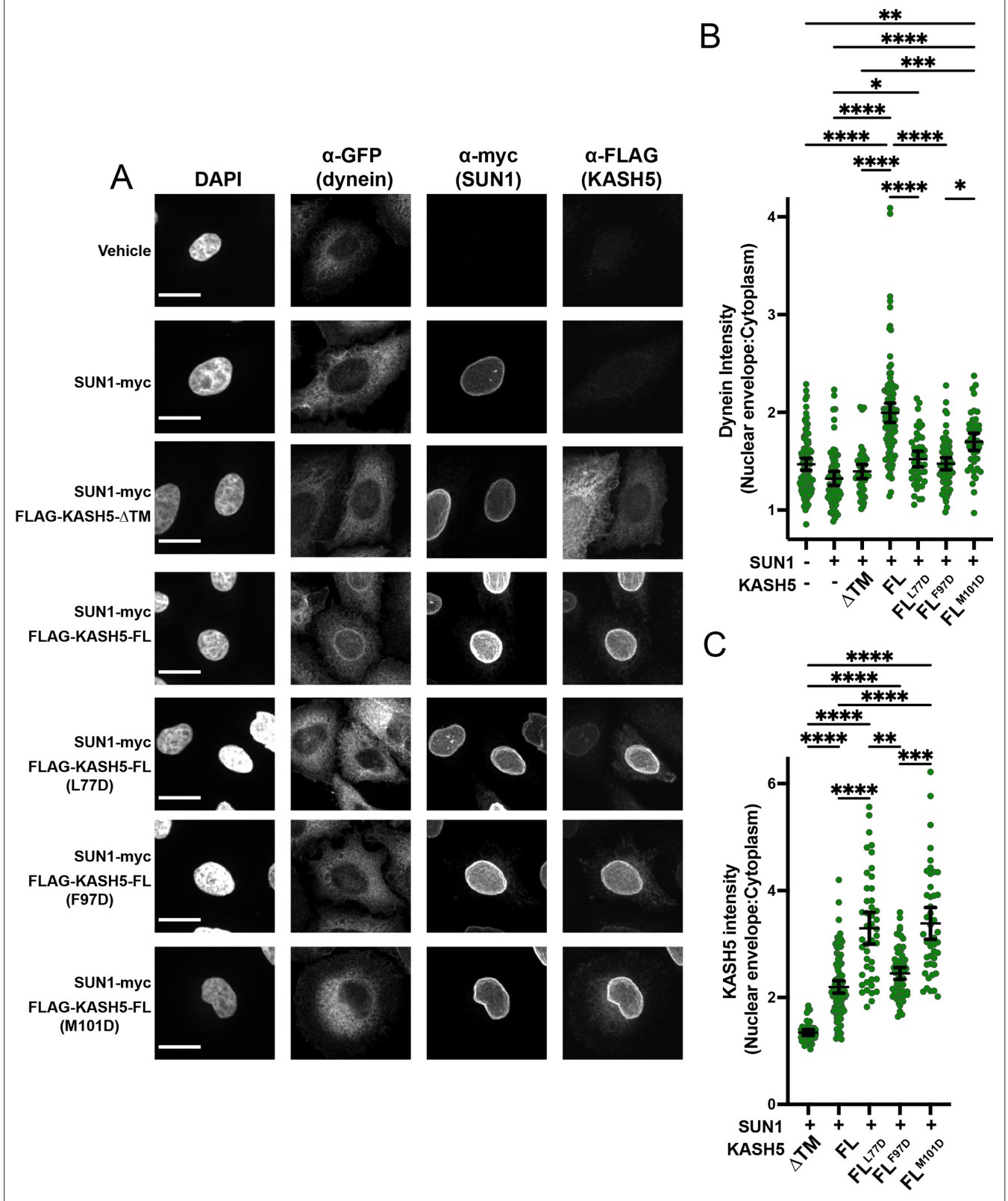

**Figure 5.** Disrupting the interaction between light-intermediate chain (LIC) and transiently expressed KASH5 inhibits dynein recruitment to the nuclear envelope (NE) of HeLa cells. (**A**) A HeLa cell line stably expressing a GFP fusion of dynein heavy chain (HC) was transfected with indicated FLAG-tagged KASH5 constructs and SUN1-myc and analyzed by immunofluorescence against GFP, FLAG, and myc. Scale bar = 20 µm. (**B and C**) Mean and 95% CI shown for the ratio of the nuclear periphery signal versus cytoplasmic signal for GFP-dynein HC (B) and FLAG-KASH5 (C). Each data point represents

*Figure 5 continued on next page*

*Figure 5 continued*

the ratio of NE: cytoplasmic intensity of dynein or KASH5 in a single cell from a total of two biological replicates. n=94, 67, 45, 102, 45, 66, and 45 for untransfected, SUN1 only, KASH5-ΔTM, KASH5-FL, KASH5-FL[L77D], KASH5-FL[F97D], and KASH5-FL[M101D]. Kruskal-Wallis test was performed for every pairwise comparison for (B) and (C). *p≤0.05; **p≤0.01; ***p≤0.001; ****p≤0.0001. Only pairwise comparisons with a p-value p≤0.05 are shown.

The online version of this article includes the following source data for figure 5:

**Source data 1.** Numerical source data relating to *Figure 5B and C*.

telomeres; *Figure 6D*) to the same extent at GFP-tagged KASH5 WT and L147D proteins. The severity of the mutations in vivo is consistent with human KASH5-FL[L77D] being the only mutant that reduced dynein and enriched KASH5 localization at the NE in HeLa cells in a statistically significant manner. To ask if the in vivo phenotypes could be enhanced by combining mutations, we generated a double mutant F167D-M171D and a triple mutant L147D-F167D-M171D. Both constructs are expressed at similar levels to GFP-KASH5 and localize to telomeres (*Figure 6—figure supplement 1A, B*). The triple mutant, but not the double mutant, decreased p150 accumulation at telomeres, suggesting that the F167D and M171D mutations do not substantially alter dynein-dynactin localization to telomeres in mouse spermatocytes (*Figure 6—figure supplement 1B, C*). The milder phenotype in vivo may be attributed to the presence of endogenous KASH5 that could homodimerize with itself or with exogenous mutant KASH5 to confer dynein binding (see Discussion for other possibilities). Together, our results suggest that human KASH5 L77 (mouse KASH5 L147) is central to KASH5-dynein binding and KASH5 recruitment to the NE.

## Discussion

We set out to discover how dynein is activated during meiosis to drive the chromosomal movements required for homolog pairing. We tested the hypothesis that KASH5 is a novel dynein activating adaptor based on its domain composition and ability to co-immunoprecipitate with dynein. Our data show that not only does KASH5's EF-hand pair bind to dynein like other activating adaptors via binding to LIC's helix 1 but also KASH5 is a bona fide activating adaptor that converts dynein and dynactin into a processive complex. We show that the interaction with KASH5 promotes dynein localization to the NE in cell culture and mouse spermatocytes. Single mutations at the KASH-dynein interface abrogate this process, highlighting the importance of this binding event in vivo. Together, our data firmly place KASH5 within the family of dynein activating adaptors and highlight how dynein is localized and activated during prophase I of meiosis.

KASH5 is unique among known activating adaptors for several reasons. First, it is the first known TM domain-containing activating adaptor. Other activating adaptors that link dynein to membranous cargos do so via a receptor protein (e.g. Rabs) associated with the cargo lipids (*Huynh and Vale, 2017*; *Horgan and McCaffrey, 2011*; *Jordens et al., 2005*). Second, KASH5 is the first known activating adaptor in the KASH (or Nesprin) family of SUN-binding proteins. Other KASH proteins, like KASH2 (also called Nesprin-2), promote dynein localization to the nucleus during nuclear migration by engaging activating adaptors like BicD2 (*Gonçalves et al., 2020*; *Tsai et al., 2020*). KASH5 directly binds dynein to form an activated dynein complex at the NE. Third, KASH5 is the only known meiosis-specific activating adaptor, consistent with the unique cargo it moves. Mammalian telomeres are generally dispersed in the nucleoplasm at steady-state in all somatic and germ cells, except in spermatocytes (or oocytes) undergoing meiosis (*Shibuya and Watanabe, 2014c*; *Palm and de Lange, 2008*). Homologous chromosome pairing in prophase I of meiosis requires the relocalization of all telomeres to the NE of spermatocytes to connect chromosomes to cytosolic dynein. Although much is known about how prophase I-specific telomere-NE attachment is established (*Pendlebury et al., 2017*; *Shibuya et al., 2015*; *Shibuya et al., 2014a*), how dynein is activated to move dynein-tethered chromosomes was not well understood. Our findings fill this gap by revealing KASH5 as the activating adaptor for chromosome motility in prophase I.

Our data show that KASH5 and other known dynein activating adaptors share a structural mechanism for recognizing dynein LIC. Interestingly, our SEC-MALS data support a model where the two copies of KASH5 EF-hand pairs from one KASH5 homodimer interact with one LIC polypeptide. We reproduced this result at higher protein concentrations (~10-fold above the $K_d$ of the KASH5 NCC-LIC complex) and in the presence of an excess in LIC, suggesting that the sub-stoichiometric amount

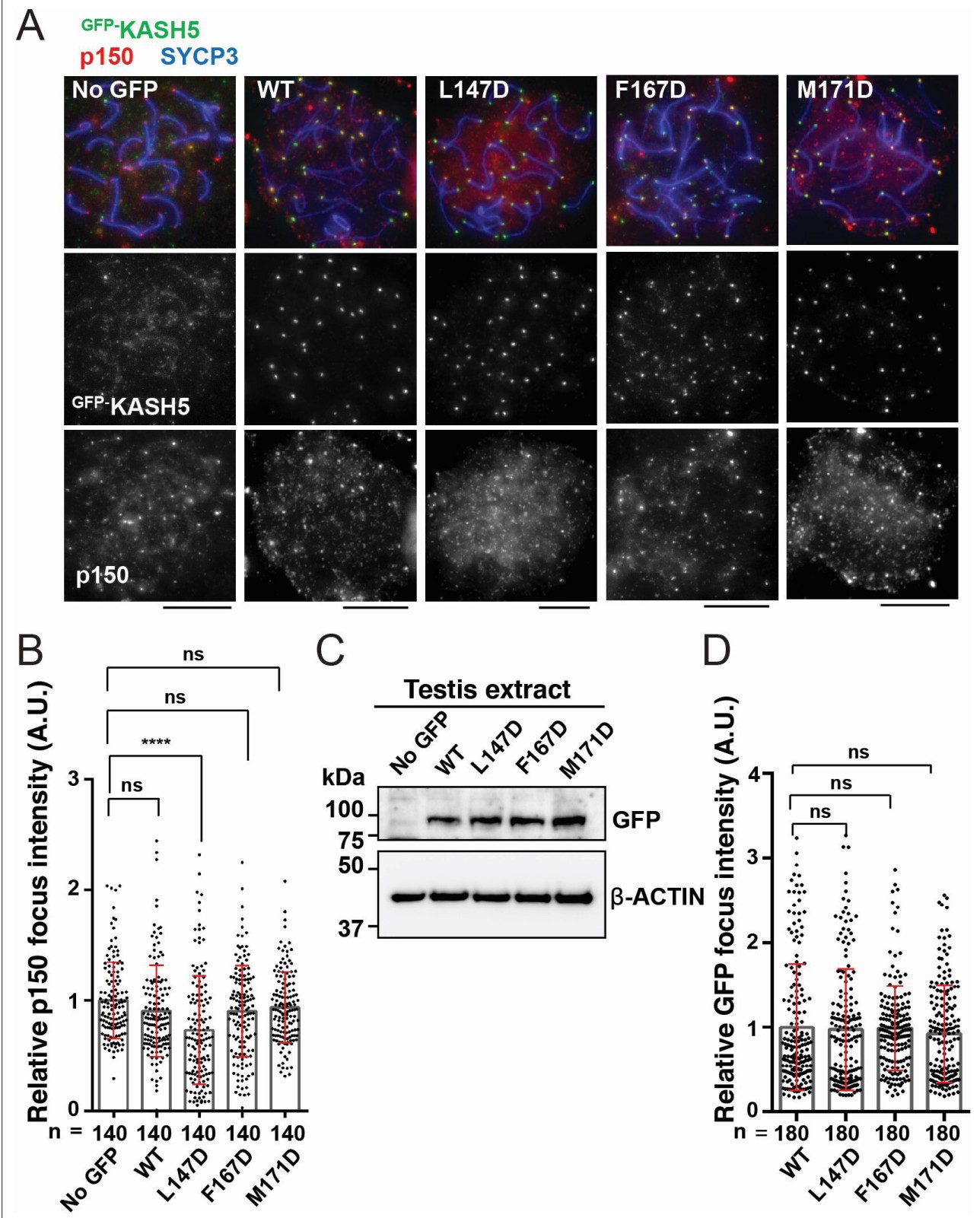

**Figure 6.** Disrupting the interaction between KASH5 and light-intermediate chain (LIC) inhibits dynactin localization at nuclear envelope-tethered telomeres in mouse spermatocytes. (**A**) Immunostaining of wild type (WT) pachytene spermatocytes expressing GFP-KASH5 by in vivo electroporation. Scale bar = 5 µm. (**B**) The quantification of p150 signal intensities normalized with the average value of GFP negative cells. Mean values with SD are shown. n shows the analyzed foci number pooled from seven cells; n=140. ns, not significant; ****p<0.0001 by Dunnett's multiple comparisons test.

*Figure 6 continued on next page*

*Figure 6 continued*

(**C**) Immunoblot of testis extracts without electroporation (no GFP) or after electroporating GFP-KASH5 WT and indicated KASH5 mutants, probed with the indicated antibodies. GFP and actin blots in each lane correspond to the same processed sample run on the same gel but blotted sequentially after stripping. (**D**) The quantification of GFP foci intensities normalized with the average value of WT. Mean values with SD are shown. n shows the analyzed foci number pooled from nine cells from electroporated mice; n=180. All analyses used Dunnett's multiple comparisons test: ns, not significant; *** p≤0.001.

The online version of this article includes the following source data and figure supplement(s) for figure 6:

**Source data 1.** Numerical source data relating to *Figure 6B and D*.

**Source data 2.** Unedited Blots relating to *Figure 6C*.

**Figure supplement 1.** Combining mutations to disrupt the interaction between KASH5 and light-intermediate chain (LIC) to inhibit dynactin localization in vivo.

**Figure supplement 1—source data 1.** Numerical source data relating to *Figure 6—figure supplement 1C*.

**Figure supplement 1—source data 2.** Unedited blots related to *Figure 6—figure supplement 1A*.

of LIC in the complex is not a result of incomplete binding. This stoichiometry is different than what has been captured in LIC helix 1 peptide-bound crystal structures of LIC-binding domains of HOOK3 and CRACR2A, which reveal a 1:1 stoichiometry (*Lee et al., 2018*; *Lee et al., 2020*). However, the HOOK3 HOOK-LIC helix 1 (PDB: 6BPH) (*Lee et al., 2018*) and the CRACR2A EF-hand pair-LIC helix 1(PDB: 6PSD) (*Lee et al., 2020*) structures were solved in the context of the N-terminal activating adaptor domains that lack the downstream CC responsible for homodimerization. It is possible that the absence of dimerization may have prevented the capture of the 2:1 stoichiometry of the activator-LIC helix complex in these structures. Interestingly, BicD2 uses a dimeric N-terminal CC-box to bind LIC. The structure of the BicD2 N-terminal CC with the LIC helix 1 (PDB: 6PSE) shows binding of two BicD2 CC helices to a single LIC helix (*Lee et al., 2020*); however, it was suggested that crystal packing may have prevented the binding of the second LIC helix to the BicD2 dimer. Indeed, a 2:2 BicD2-LIC helix interaction was inferred from ITC experiments in the same study (*Lee et al., 2020*). A major difference between these binding analyses and our SEC-MALS experiments is the use of the LIC helix-1 peptide versus full-length LIC, respectively. In this regard, a recent preprint describes a cryo-EM reconstitution of the dynein-dynactin complex including full-length LIC with another BicD family activator, BICDR1 (*Chaaban and Carter, 2022*). This study captured two BICDR1 homodimers bound to the dimeric dynein complex. While evidence from the EM maps for binding of the first LIC molecule was clear, density for the second LIC was either less resolved (for BICDR-A homodimer) or absent (for BICDR-B homodimer). Together, these results suggest that although there are two binding sites for a standalone LIC helix on dimeric activating adaptors, binding of the first full-length LIC could affect the affinity/accessibility of the second LIC polypeptide for its binding site on an activating adaptor. High-resolution structures that capture full-length LIC (and the remainder of dynein-dynactin) and dimeric activating adaptors will likely resolve these apparent differences and help determine the native stoichiometries of dynein and its activating adaptors. Determining these quaternary structural states will bear important implications for dynein velocity and force generation.

While the dimeric structure of KASH5 is consistent with its role as a dynein activating adaptor, it raises interesting questions about SUN1-KASH5 quaternary structure and its implications for chromosome motility. SUN proteins are known to trimerize (*Sosa et al., 2012*; *Zhou et al., 2012*). This creates a potential stoichiometry mismatch between SUN1 and KASH5 that can promote higher-order quaternary structures. A recent study demonstrated that binding to KASH peptides (from KASH1, KASH4, and KASH5) induces two SUN1 trimers to bind head-to-head to form a 6:6 SUN-KASH complex (*Gurusaran and Davies, 2021*). As the KASH peptide is a short (~20 aa long) monomeric region, it remains to be determined how the SUN1-KASH5 stoichiometry is affected by KASH5 dimerization. It is possible that the 6:6 geometry is retained in the native (i.e. full length) SUN1-KASH5 complex at the NE or that other higher-order structures containing multiples of SUN1 trimers and KASH5 dimers are formed in vivo. These higher-order structures might facilitate the recruitment of multiple dynein motors and the generation of large forces needed to move chromosomes across the NE.

IP-motility experiments revealed that KASH5 generated activated dynein complexes with a velocity comparable to the well-characterized activating adaptor, BicD2. However, when we reconstituted dynein-dynactin-KASH5 motility in vitro, the activated dynein complex formed less robustly and

generated complexes that moved with a significantly slower velocity unless Lis1 was present. This result suggests that Lis1 is key to forming the activated dynein-dynactin-KASH5 complexes. Furthermore, our studies suggest that Lis1 is involved in dynein's meiosis-specific function, providing new insights into an understudied area of dynein research.

We engineered nine mutations in KASH5 at the putative binding interface for helix 1 of dynein LIC. These mutants had varying effects on dynein binding with some only moderately affecting the binding between KASH5 and dynein, while others nearly abolished binding. Surprisingly, we found that the velocity of the immunoprecipitated dynein-dynactin-KASH5 complexes was comparable for all the mutants tested, regardless of their ability to bind KASH5. This result suggests that binding between activating adaptors and dynein LIC promotes the formation of the dynein-dynactin-activator complex but does not affect the velocity of the activated complex. These observations could imply that the interaction between activating adaptors and LIC might not have a role in dynein function after the activated dynein complex is formed, although this needs to be explored further for KASH5 and other known activating adaptors.

Our results confirm that KASH5's TM domain is required for robust nuclear localization of KASH5 and recruitment of dynein there, consistent with previous studies (*Horn et al., 2013*). Interestingly, KASH5 localization at the NE is dependent on SUN1 but independent of dynein binding, suggesting that it could occur before, or concomitant with, dynein recruitment to the NE in meiotic prophase I. It will be important to determine how KASH5 competes with other cytosolic dynein activators such as BicD2 in prophase I. This may be established with either downregulation of other activating adaptors in prophase I or involvement of germline-specific dynein subunit modifications that bias specificity toward KASH5. Finally, it is impressive that a single mutation at the interface between KASH5's EF-hand domain and dynein's LIC is sufficient to prevent dynein localization to the outer nuclear membrane both in HeLa cells and mouse spermatocytes transiently expressing KASH5 protein. The milder phenotype in mouse spermatocytes may be ascribed to the presence of endogenous KASH5 in these cells that can mask the mutant phenotype as already mentioned. Alternatively, or in addition, meiosis-specific proteins, protein isoforms, and post-translational modifications in dynein may partly compensate for the weakened NCC-LIC interaction. Finally, we cannot rule out the possibility that contributions of the EF-hand residues may be different in human versus mouse cells, potentially underlying a previously uncharacterized species-specificity for dynein activation in meiosis. In this regard, mouse KASH5 harbors a 70 amino acid N-terminal extension absent in other mammalian homologs, including human KASH5, and whose significance remains unknown (*Morimoto et al., 2012*). Nevertheless, our data underline the importance of the KASH5-dynein interaction for coupling telomeres to the cytoskeletal machinery in prophase I of meiosis, a prerequisite for meiotic progression and mammalian fertility.

# Materials and methods
## Cloning, plasmid construction, and mutagenesis

KASH5 constructs used in this study: FL encoded full-length human KASH5, aa 1–562; ΔTM substitutes the TM domain aa 522–542 with a six aa gly-ser linker; ΔNΔTM deletes aa 1–155 in the context of ΔTM; NCC encodes aa 1–349; CC encodes aa 156–349; and N encodes aa 1–155. All primers were purchased from IDT. Synthetic cDNA for human KASH5 and SUN1 was purchased from Horizon Discovery. Human LIC1 was cloned using the pGEX6P1-human full-length LIC1 plasmid as a template. The pGEX6P1-human full-length LIC1 plasmid was a gift from Ron Vale (Addgene plasmid # 74597; http://n2t.net/addgene:74597; RRID:Addgene_74597; *Schroeder et al., 2014*). KASH5-NCC, KASH5-N, and LIC were cloned downstream of a His-Smt3 tag in the pSmt3 vector (obtained on signing a material transfer agreement with Cornell University) for expression in the *E. coli* BL21 (DE3) strain (Novagen). KASH5-ΔTM was cloned into the pSmt3-derived vector with an N-terminal Halo tag for TIRF experiments. For expression in HEK 293T cells, KASH5 constructs (FL, ΔTM, ΔNΔTM, NCC, CC, and N) were cloned into a pcDNA3-derived vector with an N-terminal or C-terminal 3× FLAG tag. SUN1-FL cDNA was cloned into a pcDNA3-derived vector with a C-terminal 6×-myc tag. Mutations in KASH5-FL and LIC were introduced with a QuikChange site-directed mutagenesis kit (Agilent) using complementary mutant primers. The presence of the intended mutation and the absence of unintended changes in the open reading frame were verified by Sanger sequencing. For IP motility

experiments, all KASH5 constructs (FL, deletion mutants, and the point mutants) were cloned into a pcDNA3-derived vector with an N-terminal GFP tag and a C-terminal 3× FLAG tag. The pDyn1 plasmid (the pACEBac1 expression vector containing insect cell codon-optimized dynein HC [*DYNC1H1*] fused to an amino-terminal His-ZZ-TEV tag and a SNAPf tag New England Biolabs on the carboxy-terminus) and the pDyn2 plasmid (the pIDC expression vector with codon optimized *DYNC1I2*, *DYNC1LI2*, *DYNLT1*, *DYNLL1*, and *DYNLRB1*) were recombined with Cre recombinase (New England Biolabs) to generate pDyn3. PCR was used to confirm the presence of all six dynein chains. pDyn1, pDyn2, and the pFastBac plasmid with codon-optimized human full-length Lis1 (*PAFAH1B1*) with a His-ZZ-TEV tag on the amino-terminus were generous gifts from Andrew Carter (LMB-MRC, Cambridge, UK). All cloned plasmids were subjected to Sanger sequencing.

## Protein expression and purification

The His-Smt3-KASH5 (NCC and N), His-Smt3-Halo-KASH5-ΔTM, and His-Smt3-LIC proteins were expressed from the pSmt3 vector backbone in BL21 (DE3) *E. coli* cells after induction with 0.1 mM Isopropyl ß-D-1-thiogalactopyranoside (IPTG) at 25°C for 12 hr. For lysis, cell pellets were resuspended in lysis buffer 25 mM Tris-HCl (pH 7.6), 500 mM NaCl, 10 mM 2-mercaptoethanol, 0.1 mM EDTA (0.1 mM MgCl$_2$ for LIC), 1 mM PMSF, and 1× protease inhibitor (cOmplete Protease Inhibitor Cocktail, Roche) and sonicated. The lysate was clarified by centrifugation at 30,000 × g for 30 min at 4°C. The clarified supernatant was incubated with nickel-NTA agarose beads (Qiagen) for 2 hr at 4°C. Beads were collected by gravity flow and washed twice with wash buffer (25 mM Tri-HCl [pH 7.6], 150 mM NaCl, and 10 mM 2-mercaptoethanol). Beads were then washed with high-salt wash buffer (25 mM Tris-HCl [pH 7.6], 300 mM NaCl, and 10 mM 2-mercaptoethanol) and the proteins eluted from nickel-NTA agarose beads (Qiagen; using buffer containing 25 mM Tris-HCl [pH 7.6], 150 mM NaCl, and 10 mM 2-mercaptoethanol supplemented with 300 mM imidazole [pH 8]). The His-Smt3 tag was cleaved by incubating the eluted protein for 1 hr at 4°C with Ulp1 protease (1% of total eluted protein). The proteins were further subjected to SEC (Superdex 200 column for KASH5-NCC and LIC and Superdex 75 column for KASH5-N), GE Healthcare; buffer containing 25 mM Tris-HCl [pH 7.6], 100 mM NaCl, and 2 mM 1,4-Dithiothreitol (DTT). The Halo-KASH5-ΔTM was purified on an anion exchange column (MonoQ, GE Healthcare) before subjecting it to SEC (Superose 6 column, GE healthcare). For labeling purified KASH5 used in TIRF experiments, Halo-KASH5 (Promega) was mixed with a 10-fold excess of Halo-TMR for 10 min at room temperature (RT). Unconjugated dye was removed by passing the protein through Micro Bio-spin P-6 column (Bio-rad) equilibrated in GF150 buffer (25 mM HEPES [pH 7.4], 150 mM KCl, 1 mM MgCl$_2$, 5 mM DTT, and 0.1 mM Mg-ATP) supplemented with 10% glycerol. Small volume aliquots of the labeled protein were flash-frozen in liquid nitrogen and stored at –80°C.

Human dynein and human Lis1 constructs were expressed in Sf9 cells as described (*Schlager et al., 2014*; *Htet et al., 2020*; *Baumbach et al., 2017*). Briefly, pDyn3 plasmid containing the human dynein genes or the pFastBac plasmid containing full-length Lis1 was transformed into DH10EmBacY chemically competent cells with heat shock at 42°C for 15 s followed by incubation at 37°C and shaking at 220 rpm for 6 hr in S.O.C. media (Thermo Fisher Scientific). The cells were plated on LB-agar plates containing kanamycin (50 µg/ml), gentamicin (7 µg/ml), tetracycline (10 µg/ml), BluoGal (100 µg/ml), and IPTG (40 µg/ml). Cells that contained the plasmid of interest were identified with a blue/white selection after 48–72 hr. For full-length human dynein constructs, white colonies were tested for the presence of all six dynein genes with PCR. Colonies were grown overnight in LB medium containing kanamycin (50 µg/ml), gentamicin (7 µg/ml), and tetracycline (10 µg/ml) at 37°C and agitation at 220 rpm. Bacmid DNA was extracted from overnight cultures using isopropanol precipitation as described (*Zhang et al., 2017*). 2 ml of Sf9 cells in a 6-well dish at a density of 0.5 × 10$^6$ cells/ml were transfected with up to 2 µg of fresh bacmid DNA using FuGene HD transfection reagent (Promega) at a ratio of 3:1 (Fugene reagent:DNA) according to the manufacturer's directions. Cells were incubated at 27°C for 3 days without agitation. Next, the supernatant containing the virus (V0) was harvested by centrifugation (1000 × g, 5 min, 4°C). 1 ml of the V0 virus was used to transfect 50 ml of Sf9 cells at 1 × 10$^6$ cells/ml to generate the next passage of virus (V1). Cells were incubated at 27°C for 3 days with shaking at 105 rpm. Supernatant containing V1 virus was collected by centrifugation (1000 × g, 5 min, 4°C). All V1 was protected from light and stored at 4°C until further use. To express protein, 4 ml of V1 virus were used to transfect 400 ml of Sf9 cells at a density of 1 × 10$^6$ cells/ml. Cells were incubated

at 27°C for 3 days with shaking at 105 rpm and collected by centrifugation (3500 × g, 10 min, 4°C). The pellet was washed with 10 ml of ice-cold PBS and collected again via centrifugation before being flash-frozen in liquid nitrogen and stored at –80°C until needed for protein purification.

All steps for protein purification were performed at 4°C unless indicated otherwise. For dynein preparation, Sf9 cell pellets were thawed on ice and resuspended in 40 ml of dynein-lysis buffer (50 mM HEPES [pH 7.4], 100 mM sodium chloride, 1 mM DTT, 0.1 mM Mg-ATP, 0.5 mM Pefabloc, 10% [v/v] glycerol) supplemented with 1 cOmplete EDTA-free protease inhibitor cocktail tablet (Roche) per 50 ml. Cells were lysed with a Dounce homogenizer (10 strokes with a loose plunger followed by 15 strokes with a tight plunger). The lysate was clarified by centrifugation (183,960 × g, 88 min, 4°C) in a Type 70Ti rotor (Beckman). The supernatant was mixed with 2 ml of IgG Sepharose 6 Fast Flow beads (GE Healthcare Life Sciences) equilibrated in Dynein-lysis buffer and incubated for 3–4 hr with rotation along the long axis of the tube. The beads were transferred to a glass gravity column, washed with at least 200 ml of Dynein-lysis buffer and 300 ml of tobacco etch virus (TEV) buffer (50 mM Tris–HCl [pH 8.0], 250 mM potassium acetate, 2 mM magnesium acetate, 1 mM EGTA, 1 mM DTT, 0.1 mM Mg-ATP, and 10% [v/v] glycerol). For fluorescent labeling of SNAP tag, dynein-bound beads were mixed with 5 µM SNAP-Cell-TMR (New England Biolabs) for 10 min at RT. Unconjugated dye was removed with a 300 ml wash with TEV buffer at 4°C. The beads were resuspended in 15 ml of TEV buffer supplemented with 0.5 mM Pefabloc and 0.2 mg/ml TEV protease and incubated overnight with rotation along the long axis of the tube. Cleaved proteins in the supernatant were concentrated with a 100K MWCO concentrator (EMD Millipore) to 500 µl and purified via SEC on a TSKgel G4000SWXL column (TOSOH Bioscience) with GF150 buffer as the mobile phase at 1 ml/min. Peak fractions were collected, buffer exchanged into a GF150 buffer supplemented with 10% glycerol, and concentrated to 0.1–0.5 mg/ml using a 100K MWCO concentrator (EMD Millipore). Small volume aliquots were flash frozen in liquid nitrogen and stored at –80°C.

Lysis and clarification steps for the Lis1 purification were similar to the dynein purification except Lis1-lysis buffer (30 mM HEPES [pH 7.4], 50 mM potassium acetate, 2 mM magnesium acetate, 1 mM EGTA, 300 mM potassium chloride, 1 mM DTT, 0.5 mM Pefabloc, and 10% [v/v] glycerol supplemented with 1 cOmplete EDTA-free protease inhibitor cocktail tablet [Roche] per 50 ml) was used in place of dynein-lysis buffer. The clarified supernatant was mixed with 2 ml of IgG Sepharose 6 Fast Flow beads (GE Healthcare Life Sciences) and incubated for 2–3 hr with rotation along the long axis of the tube. The beads were transferred to a gravity column, washed with at least 20 ml of Lis1-lysis buffer, 200 ml of Lis1-TEV buffer (10 mM Tris–HCl [pH 8.0], 2 mM magnesium acetate, 150 mM potassium acetate, 1 mM EGTA, 1 mM DTT, and 10% [v/v] glycerol) supplemented with 100 mM potassium acetate and 0.5 mM Pefabloc; and 100 ml of Lis1-TEV buffer. TEV protease was added to the beads at a final concentration of 0.2 mg/ml, and the beads were incubated overnight with rotation along the long axis of the tube. Cleaved Lis1 in the supernatant was collected and concentrated to 500 µl with a 30K MWCO concentrator (EMD Millipore). Concentrated Lis1 was then purified via SEC on a Superose 6 Increase 10/300 GL column (Cytiva) with GF150 buffer (25 mM HEPES [pH 7.4], 150 mM KCl, 1 mM MgCl$_2$, 5 mM DTT, and 0.1 mM Mg-ATP) supplemented with 10% glycerol as the mobile phase at 1 ml/min. Peak fractions were collected, concentrated to 0.2–1 mg/ml with a 30K MWCO concentrator (EMD Millipore), frozen in liquid nitrogen, and stored at –80°C.

Dynactin was purified from HEK 293T cell lines stably expressing p62-Halo-3xFLAG as described (*Redwine et al., 2017*). Briefly, frozen pellets collected from 80 to 160 × 15 cm plates were resuspended in up to 80 ml of dynactin-lysis buffer (30 mM HEPES [pH 7.4], 50 mM potassium acetate, 2 mM magnesium acetate, 1 mM EGTA, 1 mM DTT, and 10% [v/v] glycerol) supplemented with 0.5 mM Mg-ATP, 0.2% Triton X-100, and 1 cOmplete EDTA-free protease inhibitor cocktail tablet (Roche) per 50 ml and rotated along the long axis of the tube for at least 15 min. The lysate was clarified via centrifugation (66,000 × g, 30 min, 4°C) in a Type 70 Ti rotor (Beckman). The supernatant was mixed with 1.5 ml of anti-FLAG M2 affinity gel (Sigma-Aldrich) and incubated overnight with rotation along the long axis of the tube. The beads were transferred to a glass gravity column, washed with at least 50 ml of wash buffer (dynactin-lysis buffer supplemented with 0.1 mM Mg-ATP, 0.5 mM Pefabloc, and 0.02% Triton X-100), 100 ml of wash buffer supplemented with 250 mM potassium acetate, and then washed again with 100 ml of wash buffer. 1 ml of elution buffer (wash buffer with 2 mg/ml of 3×Flag peptide) was used to elute dynactin. The eluate was collected and filtered via centrifuging through a Ultrafree-MC VV filter (EMD Millipore) in a tabletop centrifuge according to the manufacturer's

instructions. The filtered dynactin was then diluted to 2 ml in Buffer A (50 mM Tris-HCl [pH 8.0], 2 mM magnesium acetate, 1 mM EGTA, and 1 mM DTT) and loaded onto a MonoQ 5/50 GL column (Cytiva) at 1 ml/min. The column was pre-washed with 10 CV of Buffer A, 10 CV of Buffer B (50 mM Tris-HCl [pH 8.0], 2 mM magnesium acetate, 1 mM EGTA, 1 mM DTT, and 1 M potassium acetate) and then equilibrated with 10 CV of Buffer A. A linear gradient was run over 26 CV from 35 to 100% Buffer B. Pure dynactin complex eluted between 75 and 80% Buffer B. Peak fractions containing pure dynactin complex were collected, pooled, buffer exchanged into a GF150 buffer supplemented with 10% glycerol, concentrated to 0.02–0.1 mg/ml using a 100K MWCO concentrator (EMD Millipore), aliquoted into small volumes, then flash-frozen in liquid nitrogen.

## Size exclusion chromatography-multi-angle light scattering

To determine the stoichiometry and molar mass of the KASH5 constructs, LIC, and their complexes, the purified protein or the complex was loaded on a Superdex 200 10/300 size exclusion column (GE Healthcare) in line with both a DAWN HELIOS II MALS detector (Wyatt Technology) and an Optilab T-rEX differential refractometer (Wyatt Technology). Differential refractive index and light scattering data were measured, and the data analyzed using ASTRA 6 software (Wyatt Technology). Extrapolation from Zimm plots was used to calculate molecular weights using a d$n$/d$c$ value of 0.185 ml/g.

## Isothermal titration calorimetry

Thermodynamic analysis of LIC helix 1 binding to KASH5-NCC was performed by ITC using a MicroCal iTC200 (Malvern Instruments). KASH5-NCC was dialyzed for 1 day against 25 mM Tris-HCl (pH 7.6), 100 mM NaCl, and 2 mM DTT with/without 5 mM CaCl$_2$/5 mM EGTA. The LIC$^{433–458}$ peptide (purchased from GeneScript) was resuspended in the appropriate ITC buffer. 550 µM of LIC$^{433–458}$ peptide in the syringe was titrated into 50 µM of KASH5-NCC. Titrations consisted of 10 µl injections for 10 s with an interval of 300–400 s between injections at 20°C. The fitting of thermograms was done using the program Origin (OriginLab). The parameters of the fit (stoichiometry and affinity) of each experiment are provided in the figures.

## Cell culture

HeLa cells stably expressing a GFP-tagged copy of dynein HC as well as HEK 293T cells were grown at 37°C with 5% CO$_2$ in DMEM media (Corning) with 10% fetal bovine serum (FBS, Gibco) and 1% penicillin/streptomycin (Gibco). To continually select for cells expressing dynein, HeLa cells expressing GFP-dynein HC were cultured with 500 µg/ml of Geneticin (Gibco) after the first passage. While being actively cultured, all cell lines were routinely tested for mycoplasma contamination.

## Co-immunoprecipitation

HEK 293T cells were transfected with 1 µg of plasmids containing either FL (WT), various deletions (ΔTM, ΔNΔTM, NCC, CC, and N), or mutant FLAG-KASH5 constructs with Lipofectamine LTX (Invitrogen) transfection reagent using the manufacturer's instructions. At 48 hr post-transfection, the cells were washed with PBS and lysed in lysis buffer (50 mM Tris (pH 7.5), 10 mM NaCl, 2.5 mM MgCl$_2$, 1 mM DTT, 0.01% digitonin, and 1× protease inhibitor cocktail tablet [Roche]) and incubated on ice for 10 min. Lysates were passed through a 21-gauge needle 10 times and centrifuged at 16,000 × g for 10 min at 4°C. Input samples were collected and the remaining supernatant was incubated with anti-FLAG M2 affinity gel (Sigma) for 4 hr at 4°C, with rocking. After incubation, the FLAG beads were washed three times with lysis buffer, and the samples analyzed with SDS–PAGE and immunoblotting. For this, protein samples were separated on SDS–PAGE gels, transferred to a nitrocellulose membrane, and probed with the following antibodies: anti-FLAG M2-HRP (HRP, horseradish peroxidase) conjugate (Sigma, A8592; dilution 1:10,000), rabbit anti-dynein HC DYNC1H1 (Bethyl Laboratories, A304-720A-M; dilution 1:2000), rabbit anti-p150 glued dynactin (Bethyl Laboratories, A303-072A-M; dilution 1:2000), rabbit anti-dynein LIC DYNC1LI1 (Bethyl Laboratories, A304-208A-M; dilution 1:2000), and anti-rabbit HRP-conjugated secondary antibody (R&D Systems, HAF008; dilution 1:2000). Bio-rad Precision Plus Protein Dual Color Standards (in kDa from top to bottom of the gel: 250, 150, 100, 75 [pink], 50, 37, 25 [pink], 20, 15, and 10) were run on all SDS–PAGE gels, including those for western blot analysis and Coomassie-blue staining analysis.

## Single-molecule TIRF microscopy data acquisition

Single-molecule imaging was performed with an inverted microscope (Nikon, Ti2-E Eclipse) with a 100 × 1.49 N.A. oil immersion objective (Nikon, Apo). The microscope was equipped with a LUNF-XL laser launch (Nikon), with 405 nm, 488 nm, 561 nm, and 640 nm laser lines. The excitation path was filtered using an appropriate quad bandpass filter cube (Chroma). The emission path was filtered through appropriate emission filters (Chroma) located in a high-speed filter wheel (Finger Lakes Instrumentation). Emitted signals were detected on an electron-multiplying CCD camera (Andor Technology, iXon Ultra 897). Image acquisition was controlled by NIS Elements Advanced Research software (Nikon).

Single-molecule motility assays were performed in flow chambers as described (*Case et al., 1997*). No. 1-1/2 coverslips (Corning) were sonicated in 100% ethanol for 10 min and dried with air before use. Taxol-stabilized microtubules with ~10% biotin-tubulin and ~10% fluorescent-tubulin (Cytoskeleton) were prepared as described previously (*Huang et al., 2012*). Flow chambers were assembled with taxol-stabilized microtubules by incubating the following solutions interspersed with two 20 µl washes with assay buffer (30 mM HEPES [pH 7.4], 2 mM magnesium acetate, 1 mM EGTA, 10% glycerol, and 1 mM DTT supplemented with 20 µM taxol) sequentially: (1) 1 mg/ml biotin-bovine serum albumin (BSA) in assay buffer (3 min incubation), (2) 0.5 mg/ml streptavidin in assay buffer (3 min incubation), and (3) freshly diluted taxol-stabilized microtubules in assay buffer (3 min incubation). After the step where the microtubules were added, the interspersed two 20 µl washes were performed with assay buffer supplemented with 1 mg/ml casein and 20 µM taxol.

## IP-total internal reflection fluorescence

HEK 293T cells were transfected with 3 µg of KASH5-encoding plasmids (GFP-KASH5-FLAG dual-tagged) in a 10 cm dish. Transfections were performed with Lipofectamine LTX (Invitrogen). 48 hr post-transfection, media was decanted, and the cells were washed with ice-cold PBS. After centrifugation at 1000 × g for 2 min, cells were washed again with PBS and then transferred with PBS to microcentrifuge tubes for lysis. Lysis was performed with 1 ml of buffer containing 50 mM Tris-HCl, (pH 7.4), 100 mM NaCl, 0.2% Triton X-100, 1 mM DTT, 0.5 mM ATP, and 1× protease inhibitor cocktail (cOmplete, Roche) with gentle mixing at 4°C for 15 min. Lysates were then centrifuged at 16,873 × g at 4°C for 15 min. Input samples were collected, and the remaining lysate was incubated with 50 µl packed anti-FLAG M2 agarose (Sigma-Aldrich) beads for 1.5 hr at 4°C. Cells were washed twice with the lysis buffer and twice with wash buffer (30 mM HEPES pH7.4, 50 mM potassium acetate, 1 mM DTT, 0.5 mM Mg-ATP, 2 mM magnesium acetate, 1 mM EGTA, 10% glycerol, 0.01% Triton X-100, and 1× protease inhibitor cocktail [cOmplete, Roche]). Elutions were performed by incubating the washed beads with 125 µl wash buffer supplemented with 0.5 mg/ml 3×FLAG peptide at 4°C for 40 min. For imaging, the final elution was diluted such that the imaging buffer consisted of 30 mM HEPES pH7.4, 12.5 mM KOAc, 1 mM DTT, 2.75 mM Mg-ATP, 2 mM magnesium acetate, 1 mM EGTA, 10% glycerol, 0.0025% Triton X-100, 0.25× protease inhibitor cocktail (cOmplete, Roche), 15 µM Taxol, 0.75 mg/ml casein, 71.5 mM β-mercaptoethanol, 0.05 mg/ml glucose catalase, 1.2 mg/ml glucose oxidase, and 0.4% glucose. GFP-BicD2/GFP-KASH5 was imaged every 500 ms for 3 min. Two biological replicates were performed, defined as separate transfections and immunoprecipitations.

## TIRF with purified proteins

To perform single-molecule motility assays with purified proteins, dynein-dynactin-KASH5 complexes were assembled by mixing purified dynein (10–20 nM concentration) with dynactin and KASH5 at a 1:2:10 molar ratio and incubated on ice for 10 min. For experiments with only dynein and dynactin, GF150 supplemented with 10% glycerol was used in place of KASH5. Dynein-dynactin-KASH5 complexes were then incubated with Lis1 or GF150 supplemented with 10% glycerol (to buffer match for experiments without Lis1) for 10 min on ice. The mixtures of dynein, dynactin, KASH5, and Lis1 were then mixed with energy buffer and an oxygen scavenger system and flowed into the flow chamber assembled with taxol-stabilized microtubules. The final imaging buffer consisted of 28.75 mM HEPES pH 7.4, 37.5 mM KCl, 1 mM DTT, 0.25 mM MgCl$_2$, 1.5 mM magnesium acetate, 0.75 mM EGTA, 10% glycerol, 15 µM Taxol, 0.75 mg/ml casein, 71.5 mM β-mercaptoethanol, 0.05 mg/ml glucose catalase, 1.2 mg/ml glucose oxidase, 0.4% glucose, and 2.5 mM Mg-ATP. The final concentration of dynein in the flow chamber was 0.5–1 pM. The final concentration of Lis1 was 50 nM. Microtubules were imaged first by taking a single-frame snapshot. Dynein and/or KASH5-labeled with fluorophores

(TMR, Alexa647, or Alexa488) were imaged every 300ms for 3 min. At the end of acquisition, micro-tubules another snapshot was taken of the microtubule channel to assess stage drift. Movies showing significant drift were not analyzed. Each sample was imaged for less than 10 min. Two biological replicates were performed, defined as two different dynein purifications. Three technical replicates were performed.

## Single-molecule motility assay analysis

Kymographs were generated from motility movies, and dynein velocity and run length were calculated from kymographs using custom ImageJ macros as described (*Roberts et al., 2014*). Only runs longer than four frames (1.2 s) were included in the analysis. Bright protein aggregates, which were less than 5% of the population, were excluded. Velocity was reported for individual dynein complexes. The landing rate was reported per field of view by dividing the total number of events analyzed for that field of view by the sum of the length of microtubules analyzed in the field of view multiplied by the length of the movie in minutes and the concentration of dynein used in that experimental condition (only for the TIRF experiments with purified components). Percent processive events was reported per field of view by dividing the number of processive events on the microtubules analyzed per field of view by the total number of processive, diffusive, and immotile events. Data plotting and statistical analyses were performed in Prism9 (GraphPad).

## IF assay in HeLa cells

HeLa cells containing a stably integrated copy of dynein HC with a C-terminal GFP tag (*Poser et al., 2008*) were seeded in 6-well dishes at 10% confluency. After overnight incubation, DMEM media was removed and replaced with pre-warmed DMEM transfection media (with 10% FBS, but no Pen/Strep). Cells at ~20% confluency were then transfected with 0.5 µg of the SUN1-myc and FLAG-KASH5 (FL, ΔTM, or mutant FL) constructs using Lipofectamine LTX and Plus reagents (Invitrogen) in Opti-MEM (Gibco). Coverslips were washed in 100% ethanol, dried under UV light, and coated with 1× Poly-D-Lysine (Sigma-Aldrich) for 25 min at 37°C. 24 hr post-transfection, the transfected HeLa cells were recovered from the 6-well dishes with trypsin (Gibco), centrifuged, resuspended, and replated on prepared coverslips at 15% confluency, which were then allowed to acclimate overnight at 37°C with 5% $CO_2$. The transfected cells were fixed in 4% paraformaldehyde (Electron Microscopy Sciences) for 15 min at RT and washed twice with 1× PBS. Subsequently, the cells were incubated in a blocking/permeabilization buffer (0.3% Triton X-100% and 5% normal goat serum [Fisher Scientific] in 1× PBS) for 1 hr at RT and washed quickly with 1× PBS. Primary antibodies, chicken anti-GFP (1:1000, Abcam #ab13970), mouse anti-myc (1:500, Cell Signaling Technologies #9811), and rabbit anti-FLAG (1:166, ECS, Bethyl #A190-102A), were diluted in an antibody dilution buffer (0.1% Triton X-100% and 0.5% BSA, Sigma-Aldrich in 1× PBS). 60 µl of the diluted antibodies were placed on a sheet of parafilm, and the coverslips were gently flipped onto the drops and left to incubate for 2 hr at RT. The coverslips were returned to the 6-well dish and washed three times with 1× PBS before incubating them with secondary antibodies diluted in the antibody dilution buffer for 1 hr at RT in the dark. The following secondary antibodies were used: anti-chicken 488 (1:1000, Abcam #ab150173), anti-mouse 555 (1:500, Invitrogen #A11004), and anti-rabbit 647 (1:500, Invitrogen #A21244). After a 1× PBS wash step, the cells were then incubated with DAPI (1 µg/ml, Thermo Fisher) for 5 min to stain for the nucleus and washed four times with 1× PBS. Finally, the coverslips were mounted onto glass slides with Prolong Gold antifade mounting media (Thermo Cat# P36934) and cured in the dark for at least 24 hr. Slides were imaged with an inverted microscope (Nikon, Ti2-E) with a 60 × 1.49 objective (Nikon, Apo) and an X1 Spinning Disk (Yokogawa). The microscope had a LUNF-XL laser launch (Nikon), with a 405 nm, 488 nm, 561 nm, and 640 nm laser lines, and a Piezo stage (Prior). The excitation path was filtered using an appropriate quad bandpass filter cube (Chroma). The emission path was filtered through appropriate filters located in a high-speed filter wheel (Finger Lakes Instrumentation). Emitted signals were detected on a Prime 95B CMOS camera (Photometrics). Image acquisition was controlled by NIS Elements Advanced Research software (Nikon). The top and bottom of each field of view were determined by the DAPI signal, and 0.2 µm thick slices were acquired to image the entire volume. All images were acquired quantitatively with the same laser settings across experiments. Two biological replicates were performed, defined as two separate transfections.

## IF analysis in HeLa

Analysis was performed in FIJI using custom macros. Max projections of all channels were created, and the nuclei channel threshold was set to create regions of interest (ROI) around the nucleus. Only cells that were almost entirely in the field of view were analyzed. Cell boundaries were traced manually using the dynein channel as a marker for the cytoplasmic area. The nuclear values of dynein and KASH5 were defined as the mean gray value for a band of 1 μm on either side of the nuclear ROI in the respective channel. The cytoplasmic values were defined by mean gray value for the cell boundary minus the nuclear ROI expanded by 1 μm. Cells were not considered if the KASH5 mean gray value was less than 250 for cells expressing a nuclear-targeted KASH5, or if the mean gray value of SUN1 was less than 200 for cells expressing only SUN1 or KASH5-ΔTM. Cells that were transfected with KASH5-ΔTM were also only considered if they had KASH5 mean gray values for the whole cell of at least 220. All of these values were determined empirically. Data plotting and statistical analyses were performed in Prism9 (GraphPad).

## Mice

WT mice were congenic with the C57BL/6 J background. All animal experiments were approved by the Institutional Animal Care and Use Committee (#1316/18).

## Immunostaining of spermatocytes

Testis cell suspensions were prepared by mincing the tissue with flathead forceps in PBS, washing several times in PBS, and resuspending in a 1:1 mixture of PBS and hypotonic buffer (30 mM Tris [pH 7.5], 17 mM trisodium citrate, 5 mM EDTA, 2.5 mM DTT, 0.5 mM PMSF, and 50 mM sucrose). After 10 min, the sample was centrifuged, and the supernatant was aspirated. The pellet was resuspended in a 1:2 mixture of PBS and 100 mM sucrose. A total of 20 μl of fixation buffer (1% paraformaldehyde, 0.15% Triton X-100, and 1 mM sodium borate [pH 9.2 adjusted by NaOH]) was applied to a glass slide, and 3 μl of the cell suspension was added to the drop, allowed to fix for 1 hr at room temperature, and air-dried. For immunostaining, the slides were incubated with primary antibodies in PBS containing 5% BSA for 2 hr and then with Alexa Fluor 488-, 594-, or 647-conjugated secondary antibodies (1:1,000 dilution, Invitrogen) for 1 hr at room temperature. The slides were washed with PBS and mounted with VECTASHIELD medium with DAPI (Vector Laboratories).

## Exogenous protein expression in the testis

Plasmid DNA was electroporated into live mouse testes by in vivo electroporation technique (*Shibuya et al., 2014b*). Briefly, male mice at PD16–30 were anesthetized with pentobarbital, and the testes were pulled from the abdominal cavity. Plasmid DNA (10 μl of 5 μg/μl solution) was injected into each testis using glass capillaries under a stereomicroscope (M165C; Leica). Testes were held between a pair of tweezers-type electrodes (CUY21; BEX), and electric pulses were applied four times and again four times in the reverse direction at 35 V for 50 ms for each pulse. The testes were then returned to the abdominal cavity, and the abdominal wall and skin were closed with sutures. The testes were removed 24 hr after the electroporation, and immunostaining was performed. Z-stacks of each image were acquired with a 0.45 μm step size. Signal quantifications were performed with projections of Z-stack images. Signal intensities were measured using the softWoRx Data Inspector tool (Delta Vision). Telomeric regions were defined as the end of the SYCP3 signals. The neighboring background signals were measured for each data point and subtracted before quantification of intensity.

## Antibodies for mouse spermatocyte studies

The following antibodies were used: mouse antibodies against β-actin (Sigma; A2228-100UL), rabbit antibodies against GFP (Invitrogen; A11122), goat antibodies against p150 (abcam; ab11806), and chicken antibody against SYCP3 (*Zhang et al., 2019*).

## Homology modeling

The close conservation of the EF-hand pair sequences of CRACR2A and KASH5 allowed us to build a homology model of the KASH5 EF-hand pair-dynein LIC helix 1 complex. For this, we first aligned the two EF-hand pairs and then threaded KASH5 EF-hand sequences through the CRACR2A EF-hand

sequences of the CRACR2A EF-hand-dynein LIC helix 1 structure (PDB: 6PSD). No energy optimization or rotamer refinement was performed post-modeling.

## Statistical analysis

For all single-molecule velocity and run length analysis of TIRF data, significance was determined via a Kruskal-Wallis test with Dunn's multiple comparisons test. For landing rate and percent processive events analysis of TIRF data, significance was determined via a Brown-Forsythe and Welch ANOVA tests performed with a Dunnett's T3 multiple comparison test. For IF experiments where dynein and KASH5 enrichment at the NE was measured, significance was determined from a Kruskal-Wallis test with Dunn's multiple comparisons test. For quantification of signal intensities in mouse spermatocytes, significance was determined via Dunnett's multiple comparison test.

## Macro availability

All custom macros are available on GitHub, copy archived at swh:1:rev:12046f5646b717be90fe7d-04e42350d88cbce7ab (*DeSantis-Lab, 2022*).

## Acknowledgements

We thank all members of the Nandakumar and DeSantis groups for critical feedback on the manuscript. We thank Rishav Mitra and James Bardwell for help with ITC. We thank Devon Pendlebury and Varsha Venkatarangan for initial cloning of SUN1 and KASH5 cDNA, and Joanna Bird for help with protein purifications. We thank Michael Cianfrocco for technical advice and helpful discussions. This work was supported by NIH Grants, R00-GM127757 (M.D.) and R01-GM120094 (J.N.), NSF grant 2142670 (M.D.), American Cancer Society Research Scholar grant RSG-17-037-01-DMC (J.N.), European Research Council StG-801659 grant (H.S.), the Swedish Research Council 2018–03426 grant (H.S.), the Knut och Alice Wallensbergs Stiftelse KAW2019.0180 grant (H.S.), and an American Heart Association predoctoral fellowship (R.A.).

## Additional information

### Funding

| Funder | Grant reference number | Author |
| --- | --- | --- |
| National Institutes of Health | R00-GM127757 | Morgan E DeSantis |
| National Institutes of Health | R01-GM120094 | Jayakrishnan Nandakumar |
| American Heart Association | RSG-17-037-01-DMC | Jayakrishnan Nandakumar |
| European Research Council | StG-801659 | Hiroki Shibuya |
| Swedish Research Council | 2018-03426 | Hiroki Shibuya |
| Knut och Alice Wallenbergs Stiftelse | KAW2019.0180 | Hiroki Shibuya |
| American Heart Association | | Ritvija Agrawal |
| National Science Foundation | 2142670 | Morgan E DeSantis |

The funders had no role in study design, data collection and interpretation, or the decision to submit the work for publication.

### Author contributions

Ritvija Agrawal, Conceptualization, Formal analysis, Funding acquisition, Investigation, Methodology, Visualization, Writing – original draft, Writing – review and editing; John P Gillies, Formal analysis,

Investigation, Methodology, Software, Visualization, Writing – original draft, Writing – review and editing; Juliana L Zang, Jingjing Zhang, Formal analysis, Investigation, Methodology, Visualization, Writing – review and editing; Sharon R Garrott, Formal analysis, Validation, Writing – review and editing; Hiroki Shibuya, Jayakrishnan Nandakumar, Morgan E DeSantis, Conceptualization, Formal analysis, Funding acquisition, Investigation, Methodology, Project administration, Resources, Supervision, Validation, Visualization, Writing – original draft, Writing – review and editing

## Author ORCIDs
Juliana L Zang ⓘ http://orcid.org/0000-0002-5738-8355
Hiroki Shibuya ⓘ http://orcid.org/0000-0002-3400-0741
Jayakrishnan Nandakumar ⓘ http://orcid.org/0000-0001-9146-2785
Morgan E DeSantis ⓘ http://orcid.org/0000-0002-4096-8548

## Ethics
All animal experiments were approved by and performed in compliance with the regulations at the University of Gothenburg Institutional Animal Care and Use Committee (#1316/18).

## Decision letter and Author response
Decision letter https://doi.org/10.7554/eLife.78201.sa1
Author response https://doi.org/10.7554/eLife.78201.sa2

# Additional files

## Supplementary files
• Supplementary file 1. Summary of statistical information. This table contains all $K_d$ values determined via isothermal titration calorimetry (ITC); all median velocity, mean percent processivity, median run length, and mean landing rates determined via total internal reflection fluorescence (TIRF); all median intensities determined via immunofluorescence; and reports all statistical tests used and all p-values determined in the manuscript.

• Transparent reporting form

• Source data 1. Supplemental files.

## Data availability
Source data for TIRF experiments in Figure 3-6 are found in the file "Agrawal_etal_Source data" and labeled appropriately. All custom macros written for this study (used in Figure 5) are available on GitHub (https://github.com/DeSantis-Lab/Nuclear_Envelope_Localization_Macros, copy archived at swh:1:rev:12046f5646b717be90fe7d04e42350d88cbce7ab).

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
