## [Editor Report]

This manuscript identifies a meiosis-specific protein that recruits and activates the motility of the dynein-1 transport machinery at the nuclear envelope. In prophase I of meiosis, dynein moves chromosomes tethered to the nuclear envelope to expedite the search and pairing between homologous chromosomes. Previous studies have shown that dynein tethers to chromosomes via the LINC complex, which consists of a SUN protein and transmembrane KASH protein. KASH5 comprises all the known features of bona fide cargo adaptors of dynein, and this manuscript demonstrated that KASH5 directly binds dynein and activates its processive motility.

---

## [Decision Letter]

**Decision letter after peer review:**

Thank you for submitting your article "The KASH5 protein involved in meiotic chromosomal movements is a novel dynein activating adaptor" for consideration by *eLife*. Your article has been reviewed by 3 peer reviewers, including Ahmet Yildiz as the Reviewing Editor and Reviewer #1, and the evaluation has been overseen by Suzanne Pfeffer as the Senior Editor. The following individual involved in the review of your submission has agreed to reveal their identity: Richard J McKenney (Reviewer #2).

Essential revisions:

1) A major weakness of the paper is the somewhat mild phenotype observed in mouse spermatocytes expressing KASH5 mutants. As is, conclusions drawn from these experiments are not well justified and require additional experiments. While these same mutants had stronger phenotypes in non-meiotic cells, it is unclear why the phenotypes are milder in the biologically relevant cell type. The authors' explanation of endogenous wild-type KASH5 blunting these effects seems plausible, but this needs to be tested with siRNA knockdown of endogenous KASH5 while expressing the mutant protein with missense mutations. The panel also recommends the authors try combining point mutations into a single construct, which may have stronger effects on dynein binding or titration of expression to higher levels to outcompete the endogenous KASH5 protein.

2) The authors should quantify the motor run lengths, run frequencies, and the fraction of processive vs. diffusive motors in single-molecule motility assays. A visual inspection of the kymographs implies that the run lengths are substantially longer in the purified reconstituted system compared to the assays performed in cell extracts. Is this the case? The authors also claim that mutant versions exhibit very few processive runs, but the subsequent motility has the same velocity as WT. The analysis of the run frequency (the number of processive runs per micron length of a microtubule per second per μM motor/adaptor) would be needed to support this conclusion.

3) The authors should expand their discussion on the 2:1 stoichiometry of the KASH5-LIC complex. The SUN-Nesprin complexes are typically represented as trimers, and it is unclear how a trimeric complex of SUN1-KASH5 might interact with the dimeric dynein complex. The authors mention a prior study that found a similar stoichiometry in the crystal structure for the well-studied adapter proteins BicD2 and LIC1 (Lee et al. Nat. Comm. 2020). However, that study presented evidence that the 2:1 stoichiometry observed in the crystal was likely a crystal packing artifact and that BicD2 bound to two LIC1 peptides in solution. This is something the authors of the current manuscript should mention and clarify in their discussion of these results. More recently, Chaaban et al. biorXiv 2021 has shown cryo-EM evidence that the dynein dynactin complex recruits two BicDR1 adaptors. This new result might also be relevant in the Discussion.

*Reviewer #1 (Recommendations for the authors):*

In this manuscript, the authors perform a series of biochemical, biophysical, and cell biology assays to establish that KASH6 is an activator adaptor of dynein and it directly recruits dynein to the nuclear envelope. They showed that the N-terminal EF-hand domain interacts with the Light Intermediate Chain of dynein 1 at 2:1 stoichiometry and mutagenesis of the well-conserved phenylalanine residues in LIC disrupts this interaction. in vitro motility assays using cell extracts or purified proteins conclusively revealed that KASH5 is an activator adaptor of dynein. Importantly, the mutations that disrupt EF-LIC interactions reduce the frequency of processive runs, but not the speed, indicating that this interaction is important for the assembly and activation of the complex, but not required for subsequent motility. Studies in cells showed that mutations that disrupt EF-LIC interactions also reduced the recruitment of dynein to the nuclear envelope of HeLa cells, and the localization of the dynein transport machinery to telomeres across the nuclear envelope in murine spermatocytes.

Overall, the work is well done, straightforward and convincing. KASH5 is the first transmembrane dynein activating adaptor verified by in vitro motility assays. While the results are not particularly surprising, the work will be important to the researchers who study dynein, cargo transport, and meiosis, and broadly interesting to the cell biology and biophysics readership in general. I support the publication of this work in *eLife* provided that the authors can address the concerns I list below.

As is, the results presented in Figure 6 do not present a strong case of reduced localization of dynein to telomeres across the nuclear envelope when a point mutant is introduced to the EF-hand of KASH5. The authors list several reasonable possibilities to explain why they do not see a major dominant-negative effect when mutated KASH5 is overexpressed in these cells. Is it possible to knock down endogenous KASH5 with siRNA and overexpress KASH5 mutants that carry missense mutations to avoid RNAi silencing? The authors can also introduce double or triple mutants into the EF-hand to aim for a more significant effect as a single mutation may not be sufficient to fully disrupt EF-LIC interaction.

*Reviewer #2 (Recommendations for the authors):*

This is a very nice paper! I would recommend dropping the "gold standard" language from your work. This language has been pushed by Sam Reck-Peterson's lab for a few years, but I am unaware of anyone else in the field using it. Further, it is unclear to me why recombinant dynein should represent any type of "gold standard" for assays in the field. I understand that recombinant dynein seems more 'pure' than proteins pulled out of lysates, but have the authors considered if recombinant dynein is appropriately post translationally modified, as similar endogenous dynein? More importantly, the "gold standard" language implies that other assays that don't use recombinant dynein do not meet such a standard. Is this the case? Can the authors provide any examples from the literature whereby an assay that did not meet the "gold standard" provided inaccurate results to the field? Should labs that do not have the resources to produce recombinant dynein be dissuaded from experimentation in the field since they would not meet this arbitrary "gold standard"? I think the label is whimsical and exclusionary for the field and suggest the authors reconsider including it.

It is particularly ironic to use in this case, since the authors provide direct evidence here that the "gold standard" assay does NOT result in the same performance as that of motor complexes pulled out of lysates until they also include another component in the assay: LIS1. The take-home here is that "gold standard" assays comprised of purified components are only golden until future researchers realize which component(s) may have been missing from the assay. I recall when the "gold standard" force production for dynein was 1 pN, until everyone finally realized that they were assaying autoinhibited dynein. I doubt the authors want that to be their legacy.

*Reviewer #3 (Recommendations for the authors):*

1. Figure 3A-C It is puzzling that the authors quantified velocity but not run-length and some measure of event frequency (number of processive events/landing rate) that are also important parameters to establish KASH5 as a bona fide activating adaptor.

2. Figure 3H Why are there so few data points for processive events? There should also be some element of time in the processive event count.

3. Figure S3 KASH5-δ TM is not a single band. Are the shorter bands truncated off the N or C terminus?

4. Because Lis1 is involved in promoting DDK complex formation it makes sense that more processive events are observed, but why is the velocity higher?

5. Figure 4 Similar comment to earlier. Why is there no quantitation of run-length that reflects on complex stability?

6. Does KASH5 have a Spindly motif?

Figure presentation comments

Figure 1A, C The different striped patterns to identify the different domains look too similar.

Figure 3C Hard to see the mean and SD on the large dark-colored individual data points. Make data points smaller or change the color scheme. Same comment for Figure 5B, C.

---

## [Author Response]

Essential revisions:1) A major weakness of the paper is the somewhat mild phenotype observed in mouse spermatocytes expressing KASH5 mutants. As is, conclusions drawn from these experiments are not well justified and require additional experiments. While these same mutants had stronger phenotypes in non-meiotic cells, it is unclear why the phenotypes are milder in the biologically relevant cell type. The authors' explanation of endogenous wild-type KASH5 blunting these effects seems plausible, but this needs to be tested with siRNA knockdown of endogenous KASH5 while expressing the mutant protein with missense mutations. The panel also recommends the authors try combining point mutations into a single construct, which may have stronger effects on dynein binding or titration of expression to higher levels to outcompete the endogenous KASH5 protein.

The Reviewers suggested we knockdown endogenous KASH5 and/or combine KASH5 mutations to see if they enhance the in vivo phenotype. In Figure 6, we employ a highly specialized in vivo electroporation technique for introducing recombinant DNA into the testes of live mice that we have previously described (Shibuya et al., *PLoS Genet,* 2014 and Shibuya and Watanabe, *Methods Cell. Biol,* 2018). The technique is applicable for the overexpression of exogenous proteins (using cDNA) in live mice testes but not suitable for gene knockdown. The knockdown experiment in mammalian meiotic prophase cells is technically challenging because the long-term in vitro culture of mammalian meiocytes is not feasible and experiments need to be conducted within the tissue context in live mice. To our knowledge, there is only one paper where authors performed siRNA injection into live mouse testes to achieve knockdown (Dai et al., *Cell Rep*, 2017). However, the method has not been used in any follow-up study and we have not had success with it either, suggesting that it is not a practical avenue for knock down of genes in meiotic cells. Instead, and as described later, we believe a full gene knockout is more feasible.

As suggested by the Reviewers, we have overexpressed the double (F167D/M171D) and triple (L147D/F167D/M171D) mutants in WT testis by in vivo electroporation (new Figure 6- supplement). The mutant proteins were comparably expressed as the WT protein seen by the Western blot (Figure 6- supplement A). The expression of double (F167D/M171D) mutant did not affect the localization of endogenous p150, while the expression of the triple mutant dominantly impaired the localization of p150 to the same extent as the L147D single mutant (Figure 6-supplement B,C). These data suggest that L147, but not F157 or M171, is critical for the dynein interaction in a mouse meiosis-context, at least in our experimental system. The difference in the extent of the in vitro (using human KASH5 protein) and in vivo (using mouse KASH5 protein) phenotypes could be caused by multiple factors. First, key residues that mediate interaction between LIC and KASH5 may be species specific. While these residues are conserved in mice, mouse KASH5 has an extended N-terminus compared to the human (or other mammalian) KASH5 homologs. Additionally, as mentioned in the manuscript, the presence of endogenous KASH5 may hinder the dominant negative effects of F157D, M171D, or their double mutant.

To further explore the physiological significance of KASH5-dynein binding, we will perform the gene-complementation analysis of the KASH5 mutant proteins in *Kash5* knockout mouse testes, where the endogenous KASH5 is absent. Further, the generation of knock-in mice harboring the L147D, F157D, and M171D point mutations in the endogenous *Kash5* locus will be most informative to address the significance of these mutations in vivo. While we do not have these necessary mouse models (KASH5 knock out or knock-in mice) at this point, we are generating these mice for follow-up studies that are beyond the timeframe and the overall scope of the current manuscript. Finally, we would like to point out that human KASH5 L77D/mouse KASH5 L147D was the only mutation that showed statistically significant phenotypes in both HeLa cell localization assays and in mouse spermatocytes. Specifically, KASH5 L77D was more enriched at the nuclear envelope and recruited less dynein there compared to the wild type protein; the other two mutants showed statistically significant effects in either dynein or KASH5 localization at the nuclear envelope, but not both (Figure 5B and C). Thus, although all three mutations seem to impact dynein binding in vitro, only human KASH5 L77D/mouse KASH5 L147D showed robust localization phenotypes inside human and mouse cells.

2) The authors should quantify the motor run lengths, run frequencies, and the fraction of processive vs. diffusive motors in single-molecule motility assays. A visual inspection of the kymographs implies that the run lengths are substantially longer in the purified reconstituted system compared to the assays performed in cell extracts. Is this the case? The authors also claim that mutant versions exhibit very few processive runs, but the subsequent motility has the same velocity as WT. The analysis of the run frequency (the number of processive runs per micron length of a microtubule per second per μM motor/adaptor) would be needed to support this conclusion.

We thank the Editor and Reviewers for the suggestion to perform further analysis of our dynein motility data. We have now included run length, percent processivity, and run frequency (i.e. landing rate) analysis for all of the single molecule data in Figures 3 and 4 (now shown in Figure 3, 4 and Figure 3-supplement 1 and Figure 4-supplement). The additional analysis has helped further contrast the three defective mutations of KASH5 from the remaining mutants and wild type.

Depending on the comparison made, the run lengths are not necessarily longer in the pure protein TIRF experiments compared to the IP motility experiments. For example, the pure protein dynein-dynactin-KASH5-ΔTM had a median run length of 3.562 µm, while the IP-TIRF for KASH5-ΔTM had a median run length of 3.2 µm. The run length for the pure protein experiment with Lis1 included did have a longer run length than the IP-TIRF, with a median value of 4.431 µm. It is difficult to make a meaningful comparison in run lengths between these two types of experiments because they are performed in different buffers. Notably, IP-motility is performed in a final buffer that contains 0.0025% Triton X-100 and 12.5 mM KOAc, while pure protein motility is performed in a buffer containing 37.5 mM KCl and no Triton X-100.

3) The authors should expand their discussion on the 2:1 stoichiometry of the KASH5-LIC complex. The SUN-Nesprin complexes are typically represented as trimers, and it is unclear how a trimeric complex of SUN1-KASH5 might interact with the dimeric dynein complex. The authors mention a prior study that found a similar stoichiometry in the crystal structure for the well-studied adapter proteins BicD2 and LIC1 (Lee et al. Nat. Comm. 2020). However, that study presented evidence that the 2:1 stoichiometry observed in the crystal was likely a crystal packing artifact and that BicD2 bound to two LIC1 peptides in solution. This is something the authors of the current manuscript should mention and clarify in their discussion of these results. More recently, Chaaban et al. biorXiv 2021 has shown cryo-EM evidence that the dynein dynactin complex recruits two BicDR1 adaptors. This new result might also be relevant in the Discussion.

We agree with the Reviewer’s comment about the need to place our stoichiometry findings in the context of what is known for dynein and SUN-KASH complexes and detail what the physiological implications of our observations are. First, we repeated our SEC-MALS experiment with KASH5 NCC and LIC at a high concentration (~100 μM; limit of loading capacity of size-exclusion column), far above that of the K_d_ of their interaction (new data shown in Figure 2- supplement). Despite have a stoichiometric excess of LIC in the mixture analyzed, we observed a stoichiometry of 2:1, as we reported in the first submission. Thus, at least under the conditions of our SEC-MALS experiments, the ratio of KASH5 NCC:LIC is 2:1. In response to the reviewer’s request to expand on the discussion of dynein-adaptor stoichiometry as well as SUN1-KASH5 stoichiometry, we have added the following paragraphs to our revised manuscript, starting at line 376:

“Our data show that KASH5 and other known dynein activating adaptors share a structural mechanism for recognizing dynein LIC. Interestingly, our SEC-MALS data support a model where the two copies of KASH5 EF-hand pairs from one KASH5 homodimer interact with one LIC helix. We reproduced this result at higher protein concentrations (~10-fold above the Kd of the KASH5 NCC-LIC complex) and in the presence of an excess in LIC, suggesting that the sub-stoichiometric amount of LIC in the complex is not a result of incomplete binding. This stoichiometry is different than what has been captured in LIC helix 1 peptide-bound crystal structures of LIC-binding domains of HOOK3 and CRACR2A, which reveal a 1:1 stoichiometry (15,16). However, the HOOK3 HOOK-LIC helix 1 (PDB: 6BPH)(15) and the CRACR2A EF-hand pair-LIC helix 1(PDB: 6PSD) (16) structures were solved in the context of the N-terminal activating adaptor domains that lack the downstream CC responsible for homodimerization. It is possible that the absence of dimerization may have prevented the capture of the 2:1 stoichiometry of the activator-LIC helix complex in these structures. Interestingly, BicD2 uses a dimeric N-terminal CC-box to bind LIC. The structure of the BicD2 N-terminal CC with the LIC helix 1 (PDB: 6PSE) shows binding of two BicD2 CC helices to a single LIC helix(16), however, it was suggested that crystal packing may have prevented the binding of the second LIC helix to the BicD2 dimer. Indeed, a 2:2 BicD2-LIC helix interaction was inferred from ITC experiments in the same study(16). A major difference between these binding analyses and our SEC-MALS experiments is the use of the LIC helix-1 peptide versus full-length LIC, respectively. In this regard, a recent preprint describes a cryo-EM reconstitution of the dynein-dynactin complex including full-length LIC with another BicD family activator, BICDR1(50). This study captured two BICDR1 homodimers bound to the dimeric dynein complex. While evidence from the EM maps for binding of the first LIC molecule was clear, density for the second LIC was either less resolved (for BICDR-A homodimer) or absent (for BICDR-B homodimer). Together, these results suggest that although there are two binding sites for a standalone LIC helix on dimeric activating adaptors, binding of the first full-length LIC could affect the affinity/accessibility of the second LIC polypeptide for its binding site on an activating adaptor. High-resolution structures that capture full-length LIC (and the remainder of dynein-dynactin) and dimeric activating adaptors will likely resolve these apparent differences and help determine the native stoichiometries of dynein and its activating adaptors. Determining these quaternary structural states will bear important implications for dynein velocity and force generation.

While the dimeric structure of KASH5 is consistent with its role as a dynein activating adaptor, it raises interesting questions about SUN1-KASH5 quaternary structure and its implications for chromosome motility. SUN proteins are known to trimerize(51,52). This creates a potential stoichiometry mismatch between SUN1 and KASH5 that can promote higher-order quaternary structures. A recent study demonstrated that binding to KASH peptides (from KASH1, KASH4, and KASH5) induces two SUN1 trimers to bind head-to-head to form a 6:6 SUN-KASH complex(53). As the KASH peptide is a short (~20 aa long) monomeric region, it remains to be determined how the SUN1-KASH5 stoichiometry is affected by KASH5 dimerization. It is possible that the 6:6 geometry is retained in the native (i.e., full-length) SUN1-KASH5 complex at the nuclear envelope or that other higher-order structures containing multiples of SUN1 trimers and KASH5 dimers are formed in vivo. These higher-order structures might facilitate the recruitment of multiple dynein motors and the generation of large forces needed to move chromosomes across the nuclear envelope.”

Reviewer #1 (Recommendations for the authors):In this manuscript, the authors perform a series of biochemical, biophysical, and cell biology assays to establish that KASH6 is an activator adaptor of dynein and it directly recruits dynein to the nuclear envelope. They showed that the N-terminal EF-hand domain interacts with the Light Intermediate Chain of dynein 1 at 2:1 stoichiometry and mutagenesis of the well-conserved phenylalanine residues in LIC disrupts this interaction. in vitro motility assays using cell extracts or purified proteins conclusively revealed that KASH5 is an activator adaptor of dynein. Importantly, the mutations that disrupt EF-LIC interactions reduce the frequency of processive runs, but not the speed, indicating that this interaction is important for the assembly and activation of the complex, but not required for subsequent motility. Studies in cells showed that mutations that disrupt EF-LIC interactions also reduced the recruitment of dynein to the nuclear envelope of HeLa cells, and the localization of the dynein transport machinery to telomeres across the nuclear envelope in murine spermatocytes.Overall, the work is well done, straightforward and convincing. KASH5 is the first transmembrane dynein activating adaptor verified by in vitro motility assays. While the results are not particularly surprising, the work will be important to the researchers who study dynein, cargo transport, and meiosis, and broadly interesting to the cell biology and biophysics readership in general. I support the publication of this work in eLife provided that the authors can address the concerns I list below.As is, the results presented in Figure 6 do not present a strong case of reduced localization of dynein to telomeres across the nuclear envelope when a point mutant is introduced to the EF-hand of KASH5. The authors list several reasonable possibilities to explain why they do not see a major dominant-negative effect when mutated KASH5 is overexpressed in these cells. Is it possible to knock down endogenous KASH5 with siRNA and overexpress KASH5 mutants that carry missense mutations to avoid RNAi silencing? The authors can also introduce double or triple mutants into the EF-hand to aim for a more significant effect as a single mutation may not be sufficient to fully disrupt EF-LIC interaction.

Please see above our response to the Editor’s Essential revisions #1.

Reviewer #2 (Recommendations for the authors):This is a very nice paper! I would recommend dropping the "gold standard" language from your work. This language has been pushed by Sam Reck-Peterson's lab for a few years, but I am unaware of anyone else in the field using it. Further, it is unclear to me why recombinant dynein should represent any type of "gold standard" for assays in the field. I understand that recombinant dynein seems more 'pure' than proteins pulled out of lysates, but have the authors considered if recombinant dynein is appropriately post translationally modified, as similar endogenous dynein? More importantly, the "gold standard" language implies that other assays that don't use recombinant dynein do not meet such a standard. Is this the case? Can the authors provide any examples from the literature whereby an assay that did not meet the "gold standard" provided inaccurate results to the field? Should labs that do not have the resources to produce recombinant dynein be dissuaded from experimentation in the field since they would not meet this arbitrary "gold standard"? I think the label is whimsical and exclusionary for the field and suggest the authors reconsider including it.It is particularly ironic to use in this case, since the authors provide direct evidence here that the "gold standard" assay does NOT result in the same performance as that of motor complexes pulled out of lysates until they also include another component in the assay: LIS1. The take-home here is that "gold standard" assays comprised of purified components are only golden until future researchers realize which component(s) may have been missing from the assay. I recall when the "gold standard" force production for dynein was 1 pN, until everyone finally realized that they were assaying autoinhibited dynein. I doubt the authors want that to be their legacy.

We agree with the Reviewer’s concern and have accordingly removed the phrase “gold standard” from the text.

Reviewer #3 (Recommendations for the authors):1. Figure 3A-C It is puzzling that the authors quantified velocity but not run-length and some measure of event frequency (number of processive events/landing rate) that are also important parameters to establish KASH5 as a bona fide activating adaptor.

Thank you for this suggestion. We have now included a run-length analysis for all single molecule experiments (Figure 3-supplement 1 A, E)

2. Figure 3H Why are there so few data points for processive events? There should also be some element of time in the processive event count.

The number of processive events is calculated from all microtubules analyzed in a single movie. For each sample, we performed two technical replicates (two movies) from each of the two biological replicates for a total of n = 4. We find that pooling all analyzed microtubules results in the most consistent results, especially for motor complexes that have very infrequent processive events. We thank the Reviewer for pointing out that the landing rate analysis was not expressed per minute. We have corrected this in all landing rate analyses.

3. Figure S3 KASH5-δ TM is not a single band. Are the shorter bands truncated off the N or C terminus?

The Reviewer makes an excellent observation. We conclude that the truncated bands are representative of truncations occurring at the C-terminus based on the following arguments. First, we performed LC-MS on the purified KASH5-ΔTM protein to confirm that almost all the trypsin-digested peptides identified mapped to the KASH5 polypeptide (data not shown). This result rules out the shorter bands coming from other proteins in *E. coli*. Second, we did not observe such truncation products with KASH5-N or KASH5-NCC, which share their N-terminus with KASH5-ΔTM. This suggests that the truncations products are likely arising due to the sequences C-terminal to the CC in KASH5-ΔTM. Finally, secondary structure prediction (JPRED) and Alphafold prediction suggest that the sequence between the putative Spindly motif (which is immediately downstream of the CC; see discussion of Spindly motif below) and the TM domain of KASH5 is unstructured, consistent with the proteolytic susceptibility of this region during protein expression and purification.

4. Because Lis1 is involved in promoting DDK complex formation it makes sense that more processive events are observed, but why is the velocity higher?

Multiple reports have demonstrated that Lis1 increases complex formation and the velocity of moving complexes (for some references, please see Htet and Gillies et al., *Nat Cell Biol*, 2020; Elshenawy et al., *Nat. Cell Biol*, 2020; Gutierrez et al., *JCB*, 2017; Baumbach et al., *eLife*, 2017).

5. Figure 4 Similar comment to earlier. Why is there no quantitation of run-length that reflects on complex stability?

We thank the reviewer for pointing this out. We have included run-length analysis for the single molecule data shown in Figure 4 (Figure 4- supplement A)

6. Does KASH5 have a Spindly motif?

This is an excellent question. Yes, KASH5 does harbor a putative Spindly motif. We have included a short description of this in the text (beginning at line 200) and provided a new supplementary figure (Figure 3- supplement 2) to make this point. The presence and importance of the putative Spindly motif is also consistent with the NCC construct, which is truncated shortly upstream of the Spindly motif, showing slightly slower velocity, lower run frequency, and lower percent processivity than the KASH5- ΔTM (contains all soluble domains of KASH5, including the putative Spindly motif).

Figure presentation commentsFigure 1A, C The different striped patterns to identify the different domains look too similar.

We have changed the colors in Figure 1A and C to be more distinct.

Figure 3C Hard to see the mean and SD on the large dark-colored individual data points. Make data points smaller or change the color scheme. Same comment for Figure 5B, C.

We have changed the colors to make the error bars more distinct in Figure 3C and Figure 5B, C. We also increased the size of Figure 3C to increase clarity.